# APOLLO: Automated LLM and Lean Collaboration for Advanced Formal Reasoning

**Azim Ospanov** [*][†]
aospanov9@cse.cuhk.edu.hk

**Farzan Farnia** [†]
farnia@cse.cuhk.edu.hk

**Roozbeh Yousefzadeh** [*]
roozbeh.yz@gmail.com

## Abstract

Formal reasoning and automated theorem proving constitute a challenging subfield of machine learning, in which machines are tasked with proving mathematical theorems using formal languages like Lean. A formal verification system can check whether a formal proof is correct or not almost instantaneously, but generating a completely correct formal proof with large language models (LLMs) remains a formidable task. The usual approach in the literature is to prompt the LLM many times (up to several thousands) until one of the generated proofs passes the verification system. In this work, we present APOLLO (**A**utomated **PrO**of repair via **L**LM and **L**ean c**O**llaboration), a modular, model-agnostic agentic framework that combines the strengths of the Lean compiler with an LLM's reasoning abilities to achieve better proof-generation results at a low token and sampling budgets. *Apollo* directs a fully automated process in which the LLM generates proofs for theorems, a set of agents analyze the proofs, fix the syntax errors, identify the mistakes in the proofs using Lean, isolate failing sub-lemmas, utilize automated solvers, and invoke an LLM on each remaining goal with a low top-$K$ budget. The repaired sub-proofs are recombined and reverified, iterating up to a user-controlled maximum number of attempts. On the miniF2F benchmark, we establish a new state-of-the-art accuracy of 84.9% among sub 8B-parameter models (as of August 2025) while keeping the sampling budget below one hundred. Moreover, *Apollo* raises the state-of-the-art accuracy for Goedel-Prover-SFT to 65.6% while cutting sample complexity from 25,600 to a few hundred. General-purpose models (o3-mini, o4-mini) jump from 3–7% to over 40% accuracy. Our results demonstrate that targeted, compiler-guided repair of LLM outputs yields dramatic gains in both efficiency and correctness, suggesting a general paradigm for scalable automated theorem proving. The codebase is available at `https://github.com/aziksh-ospanov/APOLLO`

## 1 Introduction

Formal reasoning has emerged as one of the most challenging fields of AI with recent achievements such as AlphaProof and AlphaGeometry doing well at the International Math Olympiad competing with humans [1, 2, 3]. Formal reasoning relies both on AI models and a formal verification system that can automatically verify whether a mathematical proof is correct or not. In recent years, formal verification systems such as Lean [4] have facilitated a new form of doing mathematical research by enabling mathematicians to interact with the formal verification system and also with each other via the system, enabling larger numbers of mathematicians to collaborate with each other on a single project. As such, these formal verification systems are also called proof assistants as one can use them interactively to write a formal proof and instantaneously observe the current state of the proof and any possible errors or shortcomings in the proof generated by the compiler. Immediate access

---

[*]Huawei Hong Kong Research Center

[†]Department of Computer Science & Engineering, The Chinese University of Hong Kong

39th Conference on Neural Information Processing Systems (NeurIPS 2025).

**Common Approach: Whole-Proof Generation Pipeline**

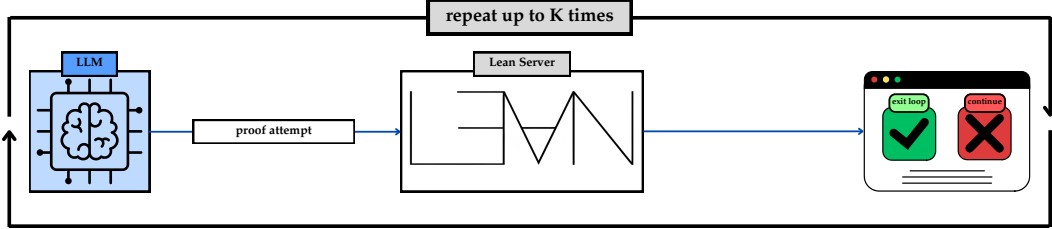

**Our Proposed Apollo Pipeline**

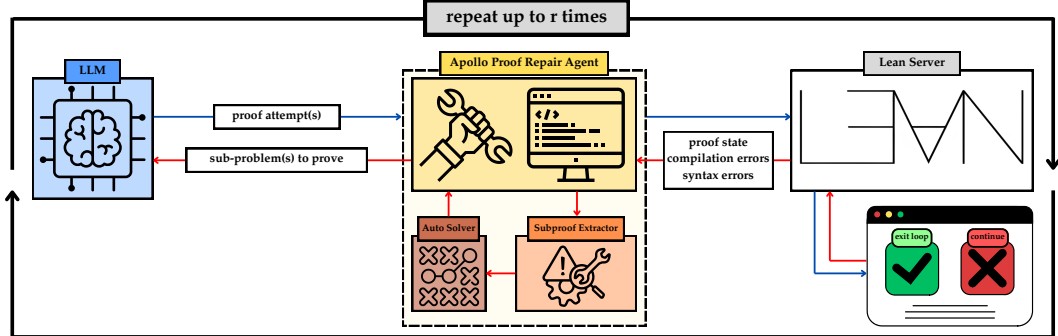

Figure 1: The summary of whole-proof generation pipeline vs. proposed Apollo agentic pipeline. LLM refers to a chosen formal theorem generator model.

to the output of Lean compiler can help a mathematician to fix possible errors in the proof. At the same time, when a proof passes the compiler with no error, other collaborators do not need to verify the correctness of the proof. This type of collaboration is transforming the field of mathematics enabling large groups of collaborators to engage in large projects of mathematical research such as the Prime Number Theorem And More project [5]. Moreover, it has given rise to the digitization of mathematics [6].

In the AI front, large language models (LLMs) are improving at mathematical reasoning in natural language, and at the same time, they have shown a remarkable ability to learn the language of those formal verification systems and write mathematical proofs in formal language. This has led to the field of Automated Theorem Proving (ATP) where the standard approach is to prompt the LLM to generate a number of candidate proofs for a given theorem which will then be verified automatically using the formal verification system. Better models, better training sets, and better training methods has led to significant advances in this field [7, 8, 9, 10, 11].

Despite these advances, the LLMs do not really get the chance to interact with the verification system the way a human does. LLMs generate many possible proofs, sometimes as many as tens of thousands, and the verification system only marks the proofs with a binary value of correct or incorrect. This common approach is illustrated in Figure 1. When the formal verification system analyzes a proof, its compiler may generate a long list of issues including syntax errors, incorrect tactics, open goals, etc. In the current approach of the community, all of those information/feedback from the compiler remains unused, as the LLM's proof generation is not directly engaged with the compiler errors of the formal verification system.

In this work, we address this issue by introducing Apollo, a fully automated system which uses a modular, model-agnostic pipeline that combines the strengths of the Lean compiler with the reasoning abilities of LLMs. Apollo directs a fully automated process in which the LLM generates proofs for theorems, a set of agents analyze the proofs, fix the syntax errors, identify the mistakes in the proofs using Lean, isolate failing sub-lemmas, utilize automated solvers, and invoke an LLM on each remaining goal with a low top-$K$ budget. The high-level overview of Apollo appears at the bottom of Figure 1.

Our contributions are as follows:

- We introduce a novel fully automated system that directs a collaboration between LLMs, Lean compiler, and automated solvers for automated theorem proving.
- We evaluate Apollo on miniF2F-test set using best LLMs for theorem proving and demonstrate that Apollo improves the baseline accuracies by significant margins while keeping the sampling costs much lower.
- We establish a new SOTA on miniF2F-test benchmark for medium-sized language models. (with parameter size 8B or less)

## 2    Related Works

**Formal theorem proving systems.** At their core, formal theorem-proving (FTP) systems employ interactive theorem provers (ITPs) to verify mathematical results. In particular, Lean4 [4] is both a functional programming language and an ITP. Each theorem, lemma, conjecture, or definition must be checked by Lean's trusted kernel, so the compiler's verdict is binary: either the statement type-checks (True) or it fails (False). This rigorous verification dramatically increases the reliability of formal mathematics; however, it also increases the complexity of proof development: authors must both comprehend the mathematical concepts and precisely encode them in the formal language. The usefulness of Lean is also shown to go beyond theorem proving, as Jiang et al. [12] showed that Lean can be adapted to natural language logical reasoning.

**Patching broken programs.** Many prior works have explored using feedback to repair broken proofs. In software engineering, "repair" typically refers to program repair, i.e. fixing code that no longer runs [13]. A common source of errors is a version mismatch that introduces bugs and prevents execution. For example, SED [14] uses a neural program-generation pipeline that iteratively repairs initial generation attempts. Other notable approaches train specialized models to fix broken code based on compiler feedback (e.g. BIFI [15], TFix [16]) or leverage unsupervised learning over large bug corpora (e.g. BugLab [17]).

**LLM collaboration with external experts.** The use of expert feedback, whether from human users or from a proof assistant's compiler, has proven effective for repairing broken proofs. Ringer [18] showcased automatic proof repair within the Coq proof assistant [19], enabling the correction of proofs in response to changes in underlying definitions. Jiang et al. [20] showed that leveraging automatic solvers with generative models yield better performance on formal math benchmarks. More recently, First et al. [21] showed that incorporating compiler feedback into LLM proof generation significantly improves the model's ability to correct errors and produce valid proofs. Another line of work [22] explored an idea of mathematician and LLM collaboration, where human experts are aided by LLMs at proof writing stage.

**Proof search methods.** Numerous recent systems use machine learning to guide the search for formal proofs. One of the methods involves using large language models with structured search strategies, e.g. Best-First Search (BFS) [23, 24, 25] or Monte Carlo Tree Search (MCTS) [26, 27, 2]. While tree-search methods reliably steer a model toward valid proofs, they result in high inference costs and often explore many suboptimal paths before success. Another line of work leverages retrieval based systems to provide context for proof generators with potentially useful lemmas. One such example is ReProver [28] that augments Lean proof generation by retrieving relevant lemmas from a proof corpus and feeding them to an LLM, enabling the model to reuse existing proofs.

**Whole proof generation.** The use of standalone LLMs for theorem proving has emerged as a major research direction in automated theorem proving. One of the earliest works presented *GPT-f* [29], a transformer-based model for theorem proving that established the LLMs ability in formal reasoning. As training frameworks and methods advance, the community has produced numerous models that generate complete proofs without external search or tree-search algorithms. Recent work shows that both supervised models [30, 31] and those trained via reinforcement learning [30, 32, 33, 34] achieve competitive performance on formal-math benchmarks. However, whole-proof generation remains vulnerable to hallucinations that can cause compilation failures even for proofs with correct reasoning chains.

**Informal Chain-of-Thought in Formal Mathematics.** Several recent works demonstrate that interleaving informal "chain-of-thought" (CoT) reasoning with formal proof steps substantially

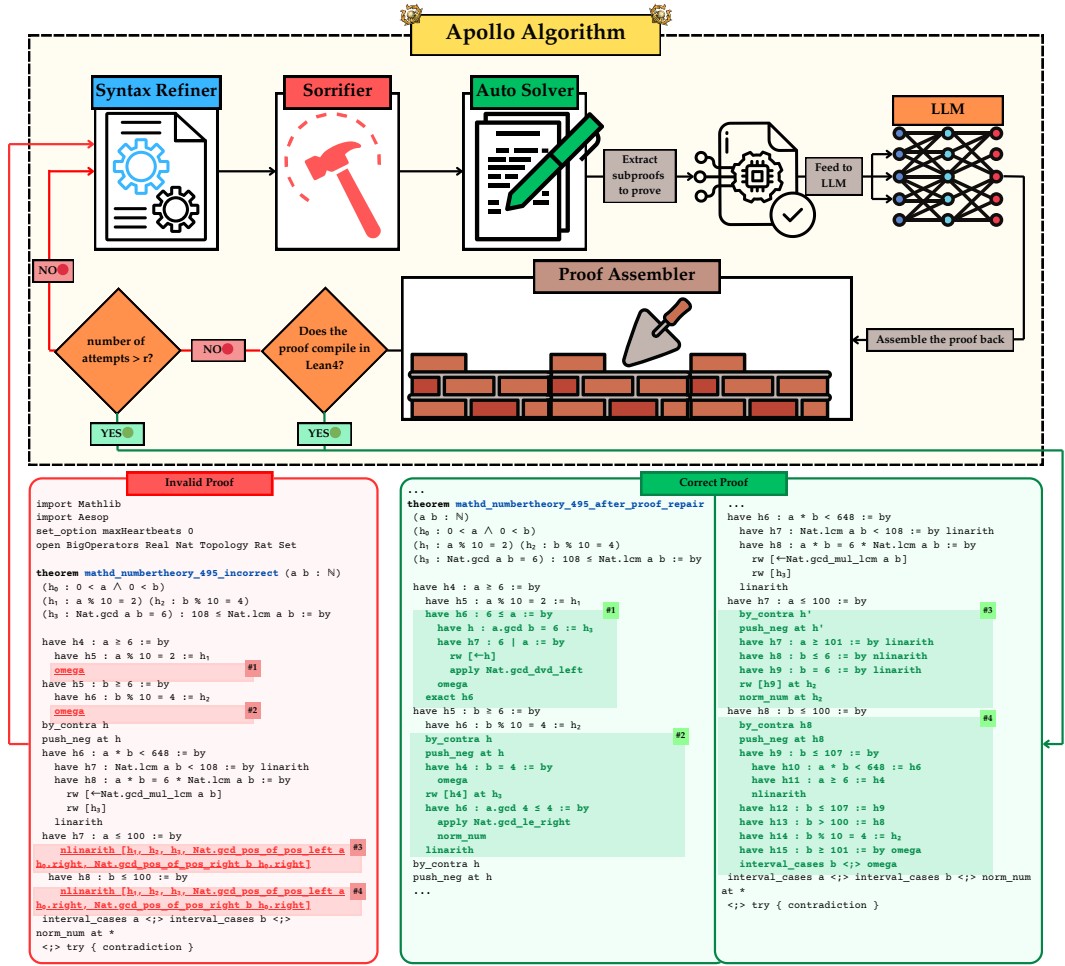

Figure 2: Overview of the Apollo framework.

improves whole-proof generation. For example, developers insert natural-language mathematical commentary between tactics, yielding large accuracy gains [31, 30, 35]. Wang, et al. [34] further show that special "thinking" tokens let the model plan its strategy and self-verify intermediate results. The "Draft, Sketch, and Prove" (DSP) [36] and LEGO-Prover [37] pipelines rely on an informal proof sketch before attempting to generate the formal proof. Collectively, these studies indicate that providing informal mathematical context, whether as comments, sketches, or dedicated tokens, guides LLMs toward correct proofs and significantly boosts proof synthesis accuracy.

## 3  Our Approach

In this section we describe Apollo, our framework for transforming an LLM's raw proof sketch into a fully verified Lean4 proof. Figure 2 illustrates the Apollo pipeline process with an attached Lean code before and after repair. We observe that often LLM is capable of producing a valid proof sketch; however, it often struggles with closing fine-grained goals. For example, statements h4, h5, h7, h8 are correct but the generated proofs for each sub-lemma throw compilation errors. Apollo identifies and fixes such proof steps and guides the LLM towards a correct solution.

Our compiler of choice is the Lean REPL [38], a "**R**ead–**E**val–**P**rint–**L**oop" for Lean4. It provides a complete list of compilation errors and warnings, each with a detailed description and the location (for example, the line number) where it occurred. Errors can include syntax mistakes, nonexistent tactics, unused tactics, and more. We aggregate this information to drive our agent's dynamic code repair and sub-problem extraction. The pseudo-code for our framework can be found in Algorithm 1 in the Appendix.

## 3.1 Syntax Refiner

The **Syntax Refiner** catches and corrects superficial compilation errors: missing commas, wrong keywords (e.g. Lean3's `from` vs. Lean4's `:= by`), misplaced brackets, etc. These syntax errors arise when a general-purpose LLM (such as o3-mini [39] or o4-mini [40]) has not been specialized on Lean4 syntax. By normalizing these common mistakes, this module ensures that subsequent stages operate on syntactically valid code. It is important to note that this module is primarily applicable to general purpose models not explicitly trained for theorem proving in Lean. In contrast, Syntax Refiner usually does not get invoked for proofs generated by LLMs trained for formal theorem proving.

## 3.2 Sorrifier

The **Sorrifier** patches any remaining compilation failures by inserting Lean's `sorry` placeholder. We first parse the failed proof into a tree of nested proof-blocks (sub-lemmas as children). Then we compile the proof with Lean REPL [38], detect the offending line or block, and apply one of three repairs:

1. *Line removal*, if a single tactic or declaration is invalid but its surrounding block may still succeed.
2. *Block removal*, if the entire sub-proof is malformed.
3. *Insert `sorry`*, if the block compiles but leaves unsolved goals open.

We repeat this procedure until the file compiles without errors. At that point, every remaining `sorry` marks a sub-lemma to be proved in later stages of Apollo. This part of the pipeline guarantees that the proof successfully compiles in Lean with warnings of presence of `sorry` statements.

Each `sorry` block corresponds to a sub-problem that the LLM failed to prove. Such blocks may not type-check for various reasons, most commonly LLM hallucination. The feedback obtained via REPL lets us to automatically catch these hallucinations, insert a `sorry` placeholder, and remove invalid proof lines.

## 3.3 Auto Solver

The **Auto Solver** targets each `sorry` block in turn. It first invokes the Lean4's `hint` to close the goal. If goals persist, it applies built-in solvers (`nlinarith`, `ring`, `simp`, etc.) wrapped in `try` to test combinations automatically. Blocks whose goals remain open stay marked with `sorry`. After applying Auto Solver, a proof may be complete with no sorry's in which case Apollo has already succeeded in fixing the proof. Otherwise, the process can repeat recursively.

## 3.4 Recursive reasoning and repair

In the case where a proof still has some remaining `sorry` statements after applying the Auto Solver, Apollo can consider each as a new lemma, i.e., extract its local context (hypotheses, definitions, and any lemmas proved so far), and recursively try to prove the lemmas by prompting the LLM and repeating the whole process of verification, syntax refining, sorrifying, and applying the Auto Solver.

At each of these recursive iterations, a lemma may be completely proved, or it may end up with an incomplete proof with one or a few `sorry` blocks. This allows the Apollo to make progress in proving the original theorem by breaking down the incomplete steps further and further until the LLM or Auto Solver can close the goals. This process can continue up to a user-specified recursion depth $r$.

## 3.5 Proof Assembler

Finally, the **Proof Assembler** splices all repaired blocks back into a single file and verifies that no `sorry` or `admit` (alias for "sorry") commands remain. If the proof still fails, the entire pipeline repeats (up to a user-specified recursion depth $r$), allowing further rounds of refinement and sub-proof generation.

Apollo's staged approach: syntax cleaning, "sorrifying," automated solving, and targeted LLM-driven repair, yields improvements in both proof-sample efficiency and final proof correctness.

# 4 Experiments

In this section, we present empirical results for Apollo on the miniF2F dataset [8], which comprises of 488 formalized problems drawn from AIME, IMO, and AMC competitions. The benchmark is evenly split into validation and test sets (244 problems each); here we report results on the miniF2F-test partition. To demonstrate Apollo's effectiveness, we evaluate it on a range of state-of-the-art whole-proof generation models. All experiments use Lean v4.17.0 and run on eight NVIDIA A5000 GPUs with 128 CPU cores. We used @32 sampling during sub-proof generation unless stated otherwise. Baseline numbers are sourced from each model's original publication works.

## 4.1 The effect of applying Apollo on top of the base models

Figure 3 shows the impact of Apollo on two state-of-the-art whole-proof generation models: Goedel Prover-SFT [31] and Kimina-Prover-Preview-Distill-7B [34]. Applying Apollo increases Goedel-Prover-SFT's top accuracy from 64.7% to 65.6% while reducing its sampling budget by two orders of magnitude (from 25,600 generated samples to only a few hundred on average). Similarly, Kimina-Prover-Preview-Distill-7B achieves a new best Kimina-Prover-Preview-Distill-7B accuracy of 75.0% with roughly one-third of the previous sample budget. We report the exact numbers in Table 1.

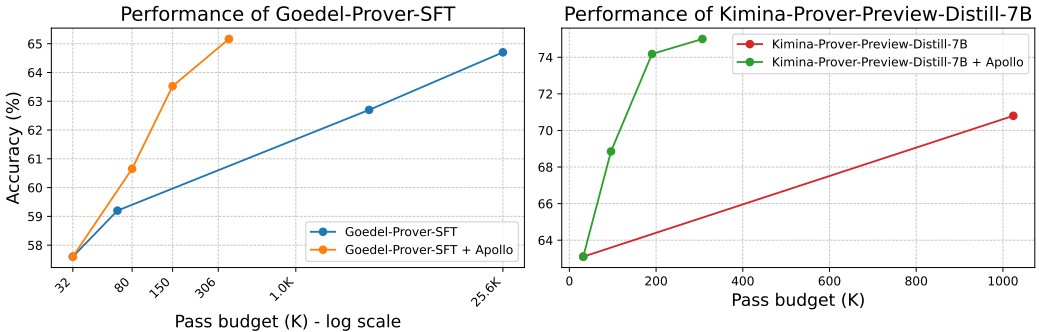

(a) Model accuracy against sampling budget

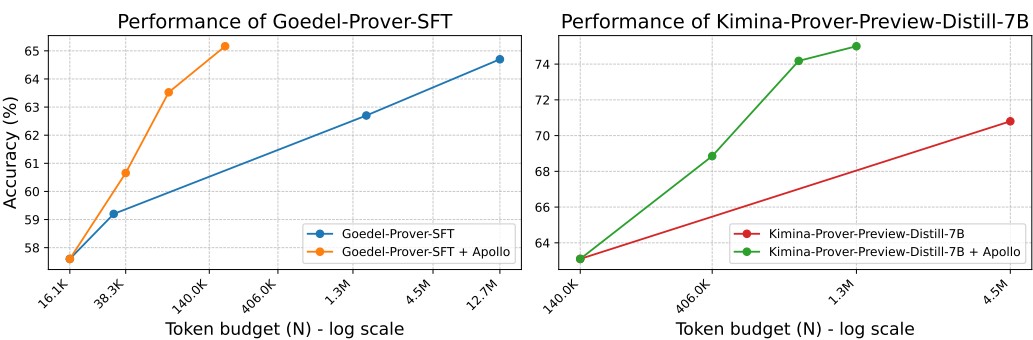

(b) Model accuracy against generated token budget

Figure 3: Performance of base models against the Apollo aided models on miniF2F-test dataset at different sample budgets and token length.

Note that Apollo's sampling budget is not fixed: it depends on the recursion depth $r$ and the LLM's ability to generate sub-lemmas. For each sub-lemma that Auto Solver fails to close, we invoke the LLM with a fixed top-32 sampling budget. Because each generated sub-lemma may spawn further sub-lemmas, the total sampling overhead grows recursively, making any single @K budget difficult to report. Sub-lemmas also tend to require far fewer tactics (and thus far fewer tokens) than the main theorem. A theorem might need 100 lines of proof while a sub-lemma might need only 1 line. Hence, sampling cost does not scale one-to-one. Therefore, to compare with other whole-proof generation

Table 1: Table results comparing sampling cost, accuracy, and token usage before and after applying Apollo. "Cost" denotes the sampling budget $K$ for whole-proof generation and the recursion depth $r$ for Apollo. Column $N$ reports the average number of tokens generated per problem in "Chain-of-Thought" mode. Bold cells highlight the best accuracy. Results are shown on the miniF2F-test dataset for two models: Kimina-Prover-Preview-Distill-7B and Goedel-SFT.

| Goedel-Prover-SFT | | | | | | Kimina-Prover-Preview-Distill-7B | | | | | |
| Base Model | | | + Apollo | | | Base Model | | | + Apollo | | |
| $K$ | $N$ | Acc. (%) | $r$ | $N$ | Acc. (%) | $K$ | $N$ | Acc. (%) | $r$ | $N$ | Acc. (%) |
|---|---|---|---|---|---|---|---|---|---|---|---|
| 32 | 16.1K | 57.6% | 0 | 16.1K | 57.6% | 32 | 140K | 63.1% | 0 | 140K | 63.1% |
| 64 | 31.8K | 59.2% | 2 | 38.3K | 60.7% | | | | 2 | 406K | 68.9% |
| 3200 | 1.6M | 62.7% | 4 | 74.6K | 63.5% | | | | 4 | 816K | 74.2% |
| 25600 | 12.7M | 64.7% | 6 | 179.0K | **65.6%** | 1024 | 4.5M | 70.8% | 6 | 1.3M | **75.0%** |

approaches, we report the average number of proof samples and token budgets. We present more in-depth analysis of inference budgets in Section B of the Appendix.

## 4.2 Comparison with SoTA LLMs

Table 2 compares whole-proof generation and tree-search methods on miniF2F-test. For each approach, we report its sampling budget, defined as the top-$K$ parameter for LLM generators or equivalent search-depth parameters for tree-search provers. Since Apollo's budget varies with recursion depth $r$, we instead report the mean number of proof sampled during procedure. Moreover, we report an average number of generated tokens as another metric for computational cost of proof generation. For some models due to inability to collect generated tokens, we leave them blank and report sampling budget only.

When Apollo is applied on top of any generator, it establishes a new best accuracy for that model. For instance, Goedel-Prover-SFT's accuracy jumps to 65.6%, surpassing the previous best result (which required 25,600 sample budget). In contrast, Apollo requires only 362 samples on average. Likewise, Kimina-Prover-Preview-Distill-7B sees a 4.2% accuracy gain while reducing its sample budget below the original 1,024. To further validate Apollo's generalizability, we tested the repair framework on Goedel-V2-8B [41], the current state-of-the-art theorem prover. We observe that, at similar sample and token budgets, Apollo achieves a 0.9% accuracy gain, whereas the base LLM requires twice the budget to reach the same accuracy. Overall, Apollo not only boosts accuracy but does so with a smaller sampling budget, setting a new state-of-the-art result among sub-8B-parameter LLMs with sampling budget of 32.

We also evaluate general-purpose LLMs (OpenAI o3-mini and o4-mini). Without Apollo, these models rarely produce valid Lean4 proofs, since they default to Lean3 syntax or introduce trivial compilation errors. Yet they have a remarkable grasp of mathematical concepts, which is evident by the proof structures they often produce. By applying Apollo's syntax corrections and solver-guided refinements, their accuracies increase from single digits (3–7%) to over 40%. These results demonstrate that in many scenarios discarding and regenerating the whole proof might be overly wasteful, and with a little help from a Lean compiler, we can achieve better accuracies while sampling less proofs from LLMs.

To further assess Apollo, we conducted experiments on PutnamBench [7], using Kimina-Prover-Preview-Distill-7B as the base model. Under whole-proof generation pipeline, LLM produced 10 valid proofs. With Apollo, the LLM produced one additional valid proof while consuming nearly half as many tokens. Results are presented in Table 3.

Overall, our results on a variety of LLMs and benchmarks demonstrate that Apollo consistently consumes fewer tokens while achieving higher accuracy, highlighting its effectiveness.

## 4.3 Distribution of Proof Lengths

We assess how Apollo affects proof complexity by examining proof-length distributions before and after repair. Here, *proof length* is defined as the total number of tactics in a proof, which serves as a proxy for proof complexity.

Table 2: Comparison of Apollo accuracy against state-of-the-art models on the miniF2F-test dataset. Token budget is computed over all generated tokens by LLM.

| Method | Model size | Sample budget | Token budget | miniF2F-test |
|---|---|---|---|---|
| *Tree Search Methods* | | | | |
| Hypertree Proof Search [26] | 0.6B | $64 \times 5000$ | - | 41.0% |
| IntLM2.5-SP+BFS+CG [24] | 7B | $256 \times 32 \times 600$ | - | 65.9% |
| HPv16+BFS+DC [42] | 7B | $600 \times 8 \times 400$ | - | 68.4% |
| BFS-Prover [25] | 7B | $2048 \times 2 \times 600$ | - | 70.8% |
| *Whole-proof Generation Methods* | | | | |
| DeepSeek-R1-Distill-Qwen-7B [43] | 7B | 32 | - | 42.6% |
| Leanabell-GD-RL [32] | 7B | 128 | - | 61.1% |
| STP [33] | 7B | 25600 | - | 67.6% |
| o3-mini [39] | - | 1 | - | 3.3% |
| | | 32 | - | 24.6% |
| o4-mini [40] | - | 1 | - | 7.0% |
| Goedel-SFT [31] | 7B | 32 | 16.1K | 57.6% |
| | | 3200 | 1.6M | 62.7% |
| | | 25600 | 12.7M | 64.7% |
| Kimina-Prover-Preview-Distill [34] | 7B | 1 | 4.4K | 52.5% |
| | | 32 | 140K | 63.1% |
| | | 1024 | 4.5M | 70.8% |
| Goedel-V2 [41] | 8B | 32 | 174K | 83.3% |
| | | 64 | 349K | 84.0% |
| | | 128 | 699K | 84.9% |
| *Whole-proof Generation Methods + Apollo* | | | | |
| o3-mini + Apollo | - | 8 | - | 40.2% (+36.9%) |
| o4-mini + Apollo | - | 15 | - | 46.7% (+39.7%) |
| Goedel-SFT + Apollo | 7B | 362 | 179K | 65.6% (+0.9%) |
| Kimina-Preview + Apollo | 7B | 307 | 1.3M | 75.0% (+4.2%) |
| Goedel-V2 + Apollo | 8B | 63 | 344K | **84.9%** (+0.9%) |

Table 3: Comparison of Apollo accuracy on the PutnamBench dataset. Token budget is computed over all generated tokens by LLM.

| Method | Model size | Sample budget | Token budget | PutnamBench |
|---|---|---|---|---|
| Kimina-Preview | 7B | 32 | 180K | 7/658 |
| Kimina-Preview | 7B | 192 | 1.1M | 10/658 |
| Kimina-Preview+Apollo | 7B | 108 | 579K | 11/658 |

Figure 4 presents violin plots for three base models: Kimina-Prover-Preview-Distill-7B, Goedel-Prover-SFT, and o4-mini. Each subplot shows two non-overlapping sets: proofs generated directly by the base model ("before") and proofs produced after applying Apollo ("after"). We only consider valid proofs verified by REPL in this setup.

The proofs that Apollo succeeds in fixing in collaboration with the LLM have considerably longer proof lengths. The mean length of proofs fixed by Apollo is longer than those written by the LLM itself at least by a factor of two in each model selection scenario. This increase indicates that the proofs which the base model alone could not prove require longer, more structured reasoning chains. By decomposing challenging goals into smaller sub-lemmas, Apollo enables the generation of these extended proofs, therefore improving overall success rate.

These results demonstrate that Apollo not only raises accuracy but also systematically guides models to construct deeper, more rigorous proof structures capable of solving harder theorems.

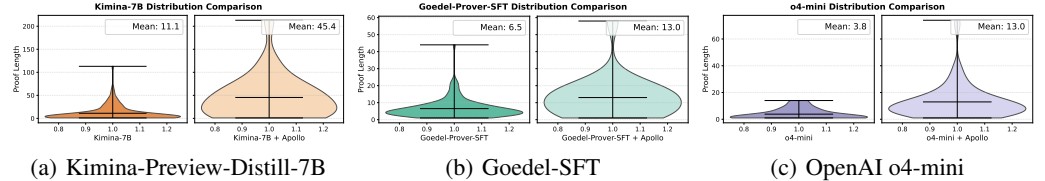

(a) Kimina-Preview-Distill-7B       (b) Goedel-SFT       (c) OpenAI o4-mini

Figure 4: Distribution of proof lengths for selected models before (left) and after (right) applying Apollo. In the "before" plots, proof lengths cover only proofs the base model proved without the help of Apollo; in the "after" plots, proof lengths cover only proofs proved with Apollo's assistance.

## 4.4 Impact of recursion depth $r$ on proof accuracy

We evaluate how the recursion-depth parameter $r$ affects accuracy on miniF2F-test. A recursion depth of $r = 0$ corresponds to the base model's standalone performance; higher $r$ values permit additional rounds of Apollo. Figure 5 plots accuracy versus $r$ for three models: o4-mini ($r = 0 \ldots 4$), Kimina-Prover-Preview-Distill-7B ($r = 0 \ldots 6$), and Goedel-Prover-SFT ($r = 0 \ldots 6$).

The o4-mini model exhibits its largest accuracy gain between $r = 0$ and $r = 1$, since its raw proof structure is often correct but contains many syntax errors and proof-step errors. Syntax Refiner fixes most of them and together with Auto Solver leads to a spike in accuracy. Beyond $r = 1$, accuracy plateaus, since o4-mini was not explicitly designed for proving formal theorems. We note that even though accuracy gains slow down drastically for o4-mini, its accuracy still increases at a slow rate, which means that pushing parameter $r$ further has a potential to increase the efficiency of o4-mini further. In contrast, both Kimina-Prover-Preview-Distill-7B and Goedel-Prover-SFT continue to improve steadily throughout different levels of recursion depth. The results suggest that decomposing the challenging problems into smaller parts allows the model to close simpler goals, slowly building towards a complete proof.

Although the total number of proof samples grows with $r$, even $r = 5$ requires only 300–400 LLM requests on average, an order of magnitude below typical budgets in the literature. Moreover, one could further increase $r$ to explore budgets in the thousands if desired.

## 4.5 Ablation Studies on different components of Apollo

To evaluate the contribution of each Apollo component, we conducted an ablation study on two base models: o4-mini (a general-purpose model) and Kimina-Prover-Preview-Distill-7B (a theorem-prover model). Results are reported in Table 4. We decomposed Apollo into three key modules: Syntax Refiner, Auto Solver, and LLM Re-invoker. The LLM invoker is the module which re-invokes the LLM to prove the sorrified sub-goals. The Sorrifier and Proof Assembler modules are excluded from this analysis, as they are essential for decomposing problems into sub-goals and assembling proofs, and thus cannot be ablated.

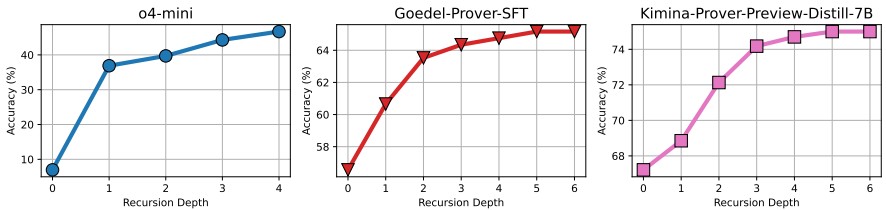

Figure 5: Performance of Apollo on miniF2F dataset at various recursion depth parameters $r$ on three different models: o4-mini, Kimina-Prover-Preview-Distill-7B and Goedel-Prover-SFT.

From the results, we observe that Syntax Refiner alone provides only a marginal benefit, and its effect is limited to the general-purpose model o4-mini; it yields no improvement on the dedicated prover. Auto Solver, when used in isolation, also produces little accuracy gain, indicating that built-in solvers in most cases do not have the capability to prove the failed sub-goals after the first round of proof generation by LLMs. However, further reasoning and proof decomposition on those sub-goals by the LLM can significantly increase the accuracy gain. The most significant accuracy gains, however, arise when Auto Solver and LLM Re-invoker are combined, which corresponds to the full Apollo framework. This demonstrates that Apollo's repair mechanism is crucial for achieving robust performance across both model types. We extend the ablation to individual module triggers during inference in Section E of the Appendix.

Table 4: Ablation study of different parts of Apollo on o4-mini (general purpose model) and Kimina-Prover-Preview-Distill-7B (theorem prover model).

| Apollo Modules | | | Model Accuracy | |
| --- | --- | --- | --- | --- |
| Syntax Refiner | AutoSolver | LLM Re-Invoker | o4-mini | Kimina-7B |
| ✗ | ✗ | ✗ | 7.0% | 63.1% |
| ✓ | ✗ | ✗ | 7.4% | 63.1% |
| ✗ | ✓ | ✗ | 7.0% | 63.5% |
| ✗ | ✗ | ✓ | 8.2% | 69.3% |
| ✓ | ✓ | ✗ | 20.5% | 63.5% |
| ✓ | ✗ | ✓ | 18.9% | 69.3% |
| ✗ | ✓ | ✓ | 10.2% | 75.0% |
| ✓ | ✓ | ✓ | 46.7% | 75.0% |

## 5 Limitations

**Integration with tree-search methods.** Our evaluation focuses exclusively on applying Apollo to whole-proof generation by LLMs and does not explore its interaction with traditional tree-search provers (e.g. BFS, MCTS). Investigating how lemma decomposition and targeted repair could accelerate or prune search paths in those systems is a promising direction for future work.

**Dependence on base model's proof sketch quality.** Apollo's ability to produce a correct formal proof largely depends on the base model and whether its initial proof has a coherent proof sketch. As [44] observe, models often tend to "cut corners," producing proofs that are much shorter than the rigorous, multi-step arguments required to generate a valid proof. When a base model fails to write a correct proof strategy (e.g. omitting critical lemmas or suggesting irrelevant tactics, or taking a wrong approach), the Apollo is less likely to repair such a proof. Enhancing the robustness of Apollo to very poor initial sketches remains an open challenge.

## 6 Conclusion

In this work, we present Apollo, a novel, modular fully automated agentic system that combines syntax cleaning, automatic solvers, and LLM-driven sub-proof generation to transform an LLM's initial proof sketch into a fully verified Lean4 proof. This framework harnesses the full power of the Lean compiler and integrated solvers, merging them with LLM systems. Applied across five whole-proof generation models, ranging from general-purpose LLMs (o3-mini, o4-mini) to specialized provers (Kimina-Prover-Preview-Distill-7B, Goedel-Prover-SFT, Goedel-V2), Apollo consistently establishes new best accuracies on the miniF2F benchmark while reducing token and sampling budgets by one to two orders of magnitude. Our empirical analysis shows that Apollo not only raises overall proof success rates but also guides models to generate longer, more structured proofs, as evidenced by a drastic increase in average successful proof length. We further demonstrate how the recursion-depth parameter $r$ trades off sample complexity against accuracy, achieving robust gains with only a few hundred samples. We believe that Apollo's collaboration between LLMs, automatic solvers, and the Lean compiler defines a paradigm of agentic systems that produce high-quality, verifiable proofs for increasingly challenging theorems.

## Acknowledgments

The work of Farzan Farnia is partially supported by a grant from the Research Grants Council of the Hong Kong Special Administrative Region, China, Project 14209920, and is partially supported by CUHK Direct Research Grants with CUHK Project No. 4055164 and 4937054. Also, the authors would like to thank the anonymous reviewers for their constructive feedback.

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

# A  Additional implementation details

Algorithm 1 presents a high-level pseudo code of the Apollo framework. We highlight that the framework is built-upon the recursive function calls up to a maximum recursion depth given by parameter $r$. **LLM** refers to a proof generator model that takes in a proof statement and outputs a formal proof attempt. **LeanCompiler** refers to Lean REPL that verifies the proof and outputs `True` or `False`. In the case of False, it also provides a detailed list of errors, their locations and types along with the error messages generated by the Lean compiler. Other functions in the pseudo code (**SyntaxRefiner**, **Sorrifier**, **AutoSolver**, **ProofAssembler**) represent the sub-modules described in the main text. To give deeper insight into Apollo's design, we now outline the rationale and responsibilities of each core module.

## A.1  `Syntax Refiner`

The **Syntax Refiner** class applies a set of rule-based corrections to eliminate syntax errors in LLM-generated Lean code. During model experimentation, we collected a corpus of common mistakes, such as stray Lean 3 keywords (`from`, `begin`...`end`), misplaced tactics, or missing delimiters, and encoded each repair as a regular-expression rule. For example:

- Convert [`from by`] to [`:= by`].
- Replace Lean 3 blocks (`begin`...`end`) with the corresponding Lean 4 structure.
- Ensure that commands like `rw` use correctly placed square brackets to avoid compilation errors.

In more complex cases, multiple regex patterns and replacements are composed to restore valid Lean 4 syntax without altering the logical content of the proof sketch.

## A.2  `Sorrifier`

The **Sorrifier** module uses a feedback loop between Apollo and the Lean REPL as follows:

1. Apollo sends the current proof candidate to REPL.
2. The REPL returns a list of error messages and their source locations.
3. Apollo parses the proof into a syntax tree, navigates to each failing sub-lemma, and attempts to fix it or replace it with a `sorry` placeholder.
4. The updated proof is re-submitted to REPL, and this cycle repeats until no error remain.

If the REPL indicates the original theorem statement itself is malformed (e.g. due to LLM hallucination), Apollo abandons the current proof and requests a fresh generation from LLM. One example of malformed formal statement is if LLM introduces a variable in a hypothesis that was not previously defined. Such case would trigger a compilation error in the formal statement; therefore, when parsing the proof, Apollo will catch this error and request a fresh proof from LLM. To guarantee consistent variable types during lemma extraction and proof assembly, Apollo sets the following Lean options to true:

```
set_option pp.instanceTypes true
set_option pp.numericTypes true
set_option pp.coercions.types true
set_option pp.letVarTypes true
set_option pp.structureInstanceTypes true
set_option pp.instanceTypes true
set_option pp.mvars.withType true
set_option pp.coercions true
set_option pp.funBinderTypes true
set_option pp.piBinderTypes true
```

so that the compiler reports explicit types (e.g. $\mathbb{Z}$ vs. $\mathbb{R}$). This fail-safe prevents trivialization of theorems and ensures correct typing in the final assembled proof.

### A.3 `AutoSolver`

The **AutoSolver** module takes the "sorrified" proof and applies a sequence of tactics to close as many goals as possible:

1. **Hint-based tactics.** It first invokes `hint`, which proposes candidate tactics. We filter out any suggestions that merely progress the goal and retain only those that fully discharge it.

2. **Built-in tactics.** If `hint` fails, AutoSolver systematically tries a suite of powerful Lean tactics, such as `nlinarith`, `norm_num`, `norm_cast`, `ring_nf`, `simp`, `simp_all`, and others, often combining them with `try` to catch failures without aborting.

3. **Recursive fallback.** Any goals that remain unsolved are reported back to Apollo. Apollo queries the REPL for the exact goal statements, spawns corresponding sub-lemmas, and then invokes the full repair loop (LLM generation, Syntax Refiner, Sorrifier, and AutoSolver) on each sub-lemma.

This tiered strategy maximizes automation while ensuring that challenging subgoals are handled via targeted recursion.

---

**Algorithm 1** Apollo Pseudocode for Formal Theorem Proving

---

**Require:** problem statement $ps$, current recursion depth $r_{current}$, maximum depth $r$
1: **if** $r_{current}$ is not initialized **then**
2:       $r_{current} \leftarrow 0$
3: **end if**
4: **if** $r_{current} > r$ **then**
5:       **return** sorry                                    ▷ If recursion depth reached, return sorry
6: **end if**
7: $proof \leftarrow$ **LLM**$(ps)$                         ▷ Generate proof from LLM
8: $proof \leftarrow$ **SyntaxRefiner**$(proof)$             ▷ Refine the proof syntax
9: $proof \leftarrow$ **Sorrifier**$(proof)$                 ▷ Patch failing sub-lemmas
10: $proof \leftarrow$ **AutoSolver**$(proof)$               ▷ Try built-in Lean solvers
11: **if** **LeanCompiler**$(proof)$ == **True then**
12:       **return** $proof$
13: **end if**
14: $r_{current} \leftarrow r_{current} + 1$
15: $n \leftarrow$ **CountSorries**$(proof)$                 ▷ Number of 'sorry' placeholders
16: $sub\_proofs \leftarrow []$
17: **for** $i = 1$ to $n$ **do**
18:       $ps_{goal} \leftarrow$ **GetTheoremGoal**$(proof, i)$
19:       $sub\_ps \leftarrow$ **TransformGoal**$(ps_{goal})$
20:       $sub\_proofs[i] \leftarrow$ **Apollo**$(sub\_ps, r_{current}, r)$
21: **end for**
22: $repaired\_proof \leftarrow$ **ProofAssembler**$(sub\_proofs)$
23: **return** $repaired\_proof$

---

## B    Discussion on maximum budget of @K and Apollo

To better highlight Apollo's contribution to token efficiency, we report token counts under two settings: (1) the budget over all attempted miniF2F-test problems, and (2) the budget restricted to proofs where Apollo was invoked. For instance, if the base LLM proves a theorem without Apollo's assistance, its token usage is included only in the first setting. In contrast, if the base LLM cannot prove the theorem on its own but succeeds with Apollo's help, its token usage is counted in the second setting. Results for setting (1) are shown in Table 5, while results for setting (2) appear in Table 6.

We emphasize that, regardless of the token-count setting, Apollo consistently consumes fewer tokens while achieving higher accuracy, highlighting its effectiveness. Throughout this work, we primarily report results under setting (2), as it more directly highlights Apollo's contribution: the reported token usage is not artificially reduced by problems that the base LLM could solve independently. This makes Scenario (2) a more representative measure of our contribution.

Table 5: Comparison of maximum/average sample/token budget across @$K$ sampling and Apollo repair on the entire miniF2F-test dataset.

| Model | Max sample budget | Average sample budget | Max token budget | Average token budget |
|---|---|---|---|---|
| Goedel-SFT | 25600 | 9730 | 12.7M | 4.8M |
| Goedel-SFT+Apollo | 1100 | 335 | 548K | 167K |
| Kimina-7B | 1024 | 373 | 1.3M | 1.6M |
| Kimina-7B+Apollo | 888 | 189 | 3.8M | 818K |

Table 6: Comparison of maximum/average sample/token budget across @$K$ sampling on the entire miniF2F-test dataset and Apollo repair exclusively on Apollo-assisted problems.

| Model | Max sample budget | Average sample budget | Max token budget | Average token budget |
|---|---|---|---|---|
| Goedel-SFT | 25600 | 9730 | 12.7M | 4.8M |
| Goedel-SFT+Apollo | 1100 | 618 | 548K | 307K |
| Kimina-7B | 1024 | 373 | 1.3M | 1.6M |
| Kimina-7B+Apollo | 888 | 367 | 3.8M | 1.6M |

## C  Additional evaluation results

We also conducted experiments on the ProofNet [9] dataset using the Kimina-Preview-Prover-Distill-7B model. Table 7 summarizes the sample/token budget and accuracy. Incorporating the Apollo agent with Kimina not only improves accuracy by 7% but also reduces token consumption by 25%.

Moreover, we observe rising adoption of REPL-based repair, introduced in [44]. Currently, models such as Goedel-Prover-V2 are fine-tuned on REPL outputs paired with faulty Lean code, enabling the model to repair missing or incorrect proof fragments using this feedback. Following [44], we compare Apollo against a REPL-feedback baseline using the general-purpose o3-mini model. Table 8 presents the results: Apollo outperforms the REPL-feedback approach while requiring a smaller sample budget. We view REPL-based feedback as complementary and expect it to work well in conjunction with recursive, compiler-aided repair of Lean proofs.

---

**Feedback prompt schema**

This is an incorrect proof:
[model's last proof]
Compilation errors are as follows:
[Lean's error messages]
Based on this feedback, produce a correct raw Lean code for the following problem:
[header]
[informal prefix]
[formal statement]

---

We also evaluated Apollo against the RMaxTS method proposed in [45]. This approach employs Monte Carlo Tree Simulation to explore the proof space and truncates the search whenever Lean compiler errors are encountered. RMaxTS has been shown to further improve the accuracy of whole-generation provers. For our evaluation, we used Kimina-Prover-Preview-Distill-7B as the base model. Table 9 reports the results. At a comparable sample budget of approximately 2M tokens, Apollo outperforms RMaxTS by 10.7%. These findings further reinforce the effectiveness of the agentic recursive repair approach.

Table 7: Comparison of Apollo accuracy on the ProofNet dataset. Token budget is computed over all generated tokens by LLM.

| Method | Model size | Sample budget | Token budget | ProofNet |
|---|---|---|---|---|
| Kimina-Preview | 7B | 128 | 559K | 11.3% |
| Kimina-Preview+Apollo | 7B | 96 | 415K | 18.3% |

Table 8: Comparison of Apollo accuracy against compiler (REPL) feedback based repair on the miniF2F-test dataset. Token budget is omitted since OpenAI API does not provide exact token budget with its response.

| Method | Sample budget | Accuracy |
|---|---|---|
| o3-mini | 1 | 3.3% |
| o3-mini | 32 | 24.6% |
| o3-mini + REPL feedback repair | 5 | 17.2% |
| o3-mini + REPL feedback repair | 10 | 25.4% |
| o3-mini + Apollo | 8 | 40.2% |

## D Computational resource considerations

Our emphasis on reducing sample complexity is driven by resource allocation. CPU performance is not a limiting factor in our setup: the Lean kernel remains efficient even under high parallelization, with proof verification typically completing in 6–200 seconds. To guard against inefficient structures, we impose a 5-minute compilation timeout, which is rarely approached in practice.

By contrast, GPU resources were the primary bottleneck. Running Kimina-Prover-Preview-Distill-7B on a single A5000 GPU with a 16,384-token context window takes 700–2,000 seconds per problem, over 11.5 minutes at @32 sampling. Increased sampling exacerbates this cost, and even with eight GPUs throughput remains difficult to scale. While Lean-based servers such as Kimina-server continue to improve verification efficiency through CPU parallelization, the growing overhead of sampling highlights the need to minimize LLM invocations alongside improving theorem-proving accuracy.

Table 9: Comparison of Apollo accuracy against other sampling methods on the miniF2F-test dataset. Token budget is computed over all generated tokens by LLM.

| Method | Model size | Sample budget | Token budget | Accuracy |
|---|---|---|---|---|
| Kimina-Preview | 7B | 1024 | 4.5M | 70.8% |
| Kimina-Preview+RMaxTS [45] | 7B | $4 \times 128$ | 2.0M | 64.3% |
| Kimina-Preview+Apollo | 7B | 589 | 2.2M | 75.0% |

# E    Extension of ablation study

To further elaborate on the ablation results presented in the main text, we measured the number of relevant Apollo module calls during inference. The results are shown in Table 10. Specifically, we counted the number of invocations for each Apollo-assisted problem in the miniF2F-test subset. The findings are consistent with the ablation study reported in the main text.

Table 10: Number of Apollo module triggers across different models

| Model | Syntax Refiner | Auto Solver | LLM Re-Invoker |
|---|---|---|---|
| Goedel-SFT | 0.0% | 100.0% | 100.0% |
| Kimina-7B | 0.0% | 100.0% | 100.0% |
| o3-mini | 100.0% | 100.0% | 70.0% |
| o4-mini | 94.7% | 100.0% | 64.9% |

# F    An example of Apollo proof repair procedure

Figure 6 shows a detailed run of the Apollo framework on `mathd_algebra_332`. First, the LLM generates an initial proof sketch, which fails to type-check in Lean. Apollo then applies its `Syntax Refiner` to correct simple errors (e.g. changing "from by" to ":= by") and to strip out all comment blocks. Next, the `Sorrifier` replaces each failing sub-lemma with a `sorry` statement (six in this example). The `AutoSolver` module then attempts to discharge these sub-lemmas: it resolves four of them, leaving lemmas #5 and #6 unsolved. Apollo recurses on those two: each is passed again through LLM, `Syntax Refiner`, `Sorrifier`, and `AutoSolver`. After this single recursive iteration, all goals are closed. Finally, Apollo reassembles the full proof and returns it to the user.

This example illustrates Apollo's repair logic. In this case the proof was fixed in one iteration; more complex theorems may require deeper recursive repair.

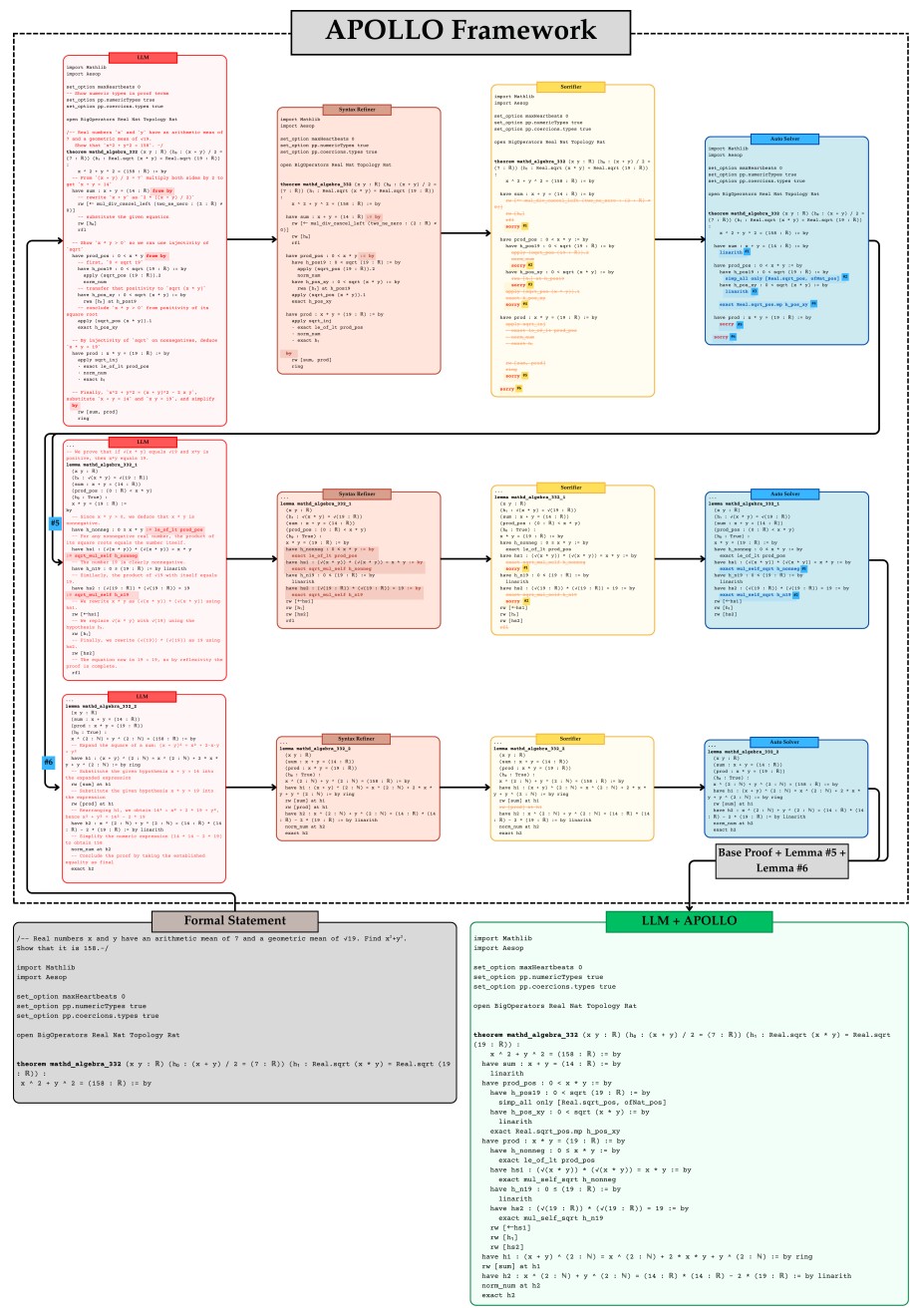

Figure 6: Step-by-step Apollo repair on `mathd_algebra_332`. The initial proof and all sub-lemma proofs are generated with o4-mini, then systematically repaired and reassembled by Apollo.

Figure 7: Proof attempt of `mathd_algebra_158` produced by the base model (o4-mini) versus after Apollo's targeted repair. The left shows the original end-to-end LLM output, which does not compile in Lean; the right shows the corrected, repaired proof assembled by Apollo that closes all of the goals.

# G Comparison between base proofs generated from LLMs and Apollo-repaired counterparts

In this section, we present and discuss examples of initial proof attempts alongside their Apollo-repaired counterparts.

Figures 7, 8, 9 illustrate cases where Apollo repairs the proof without invoking the LLM again. In each example, the LLM struggles to generate the precise tactics needed to close certain goals; however, we find that regenerating the entire proof is unnecessary. Instead, built-in Lean solvers can discharge those subgoals directly. Moreover, in Figure 7, problematic block #3 is resolved simply by removing it, since the surrounding proof context is correct. Thus, omitting problematic lines can sometimes yield a valid proof.

In Figure 10, one goal is expanded via an LLM query, but the model injects extra tactics that trigger a `no goals to be solved` error. Apollo then repairs the first block using a combination of LLM generation and AutoSolver, while the second block is removed entirely.

Figure 11 illustrates a case where the model fails to apply `nlinarith` to discharge the goal. We observe that, when uncertain, the model often resorts to broad tactics in hopes of closing the goal. Here, the goal is too complex for `nlinarith`, so Apollo leverages both the LLM and AutoSolver to guide the proof to completion.

Figure 12 illustrates an example of proof that required a recursion depth $r = 5$ to repair the initial proof attempt. We observe that LLM's proof structure is correct; however, in many cases it over-relies on built-in solvers and rewrites. However, since the goals are too complex, this approach leads to a total of nine error blocks. Repairing those blocks requires aid from both AutoSolver module of

Apollo and LLM itself. We show that if LLM grasps the correct approach, then Apollo is able to repair fine grained logic errors to produce a correct proof.

Figure 13 illustrates a case where the LLM produces an incomplete proof sketch. The first half of the proof is correct, but the model fails to discharge the remaining goal and instead uses `all_goals positivity`. Apollo detects this error and performs two additional recursive repair iterations to complete the proof.

Figure 14 illustrates a case where the base model takes a lazy approach by trying to close all goals with `linarith`. In contrast, Apollo's repaired proof performs additional setup and discharges each goal by first proving the necessary auxiliary hypotheses. This example shows that, although the model often understands the high-level strategy, it lacks the fine-grained tactics (and compiler feedback) needed to close subgoals. Apollo decomposes the proof, identifies the failure points, and applies targeted repairs to generate a fully verified solution.

Figure 15 shows another scenario in which Apollo successfully closes the first two blocks with `linarith`, but the final block requires a deeper reasoning chain. Apollo augments the proof by introducing and proving the missing lemmas, which leads to a correct solution with a series of rewrites.

Figure 16 presents an example of a very large repair. As in Figure 12, Apollo requires $r = 5$ to fully fix the proof. The final proof length increases from 100 lines to 216, which means Apollo applied roughly $\times 2.2$ more tactics to close the goal. The repair relied on combined effort from AutoSolver and the LLM to close all goals. We show that, even for large, complex proofs, Apollo can repair each failing sub-block and generate a correct solution.

Figure 8: Proof attempt of `mathd_algebra_141` produced by the base model (o4-mini) versus after Apollo's targeted repair. The left shows the original end-to-end LLM output, which does not compile in Lean; the right shows the corrected, repaired proof assembled by Apollo that closes all of the goals.

**LLM**

```
import Mathlib
import Aesop

set_option maxHeartbeats 0

open BigOperators Real Nat Topology Rat

theorem imo_1983_p6_llm (a b c : ℝ) (h₀ : 0 < a ∧ 0 <
b ∧ 0 < c) (h₁ : c < a + b) (h₂ : b < a + c)
    (h₃ : a < b + c) : 0 ≤ a ^ 2 * b * (a - b) + b ^ 2
* c * (b - c) + c ^ 2 * a * (c - a) := by

  nlinarith [sq_nonneg (a - b), sq_nonneg (b - c),   #1
    sq_nonneg (c - a), sq_nonneg (a - b),
    sq_nonneg (b - c), sq_nonneg (c - a),
    mul_pos h₀.left h₀.right.left,
    mul_pos h₀.left h₀.right.right,
    mul_pos h₀.right.left h₀.right.right]
```

**LLM + APOLLO**

```
import Mathlib
import Aesop

set_option maxHeartbeats 0

open BigOperators Real Nat Topology Rat

theorem imo_1983_p6_apollo (a b c : ℝ) (h₀ : 0 < a
∧ 0 < b ∧ 0 < c) (h₁ : c < a + b) (h₂ : b < a + c)
    (h₃ : a < b + c) : 0 ≤ a ^ 2 * b * (a - b) + b ^
2 * c * (b - c) + c ^ 2 * a * (c - a) := by

  nlinarith [sq_nonneg (a - b), sq_nonneg (b - c),   #1
    sq_nonneg (c - a),
    mul_pos (sub_pos.mpr h₁)
    (sub_pos.mpr h₂), mul_pos
    (sub_pos.mpr h₂)
    (sub_pos.mpr h₃), mul_pos (sub_pos.mpr h₃
    (sub_pos.mpr h₁)]
```

Figure 9: Proof attempt of `imo_1983_p6` produced by the base model (Kimina-Prover-Preview-Distill-7B) versus after Apollo's targeted repair. The left shows the original end-to-end LLM output, which does not compile in Lean; the right shows the corrected, repaired proof assembled by Apollo that closes all of the goals.

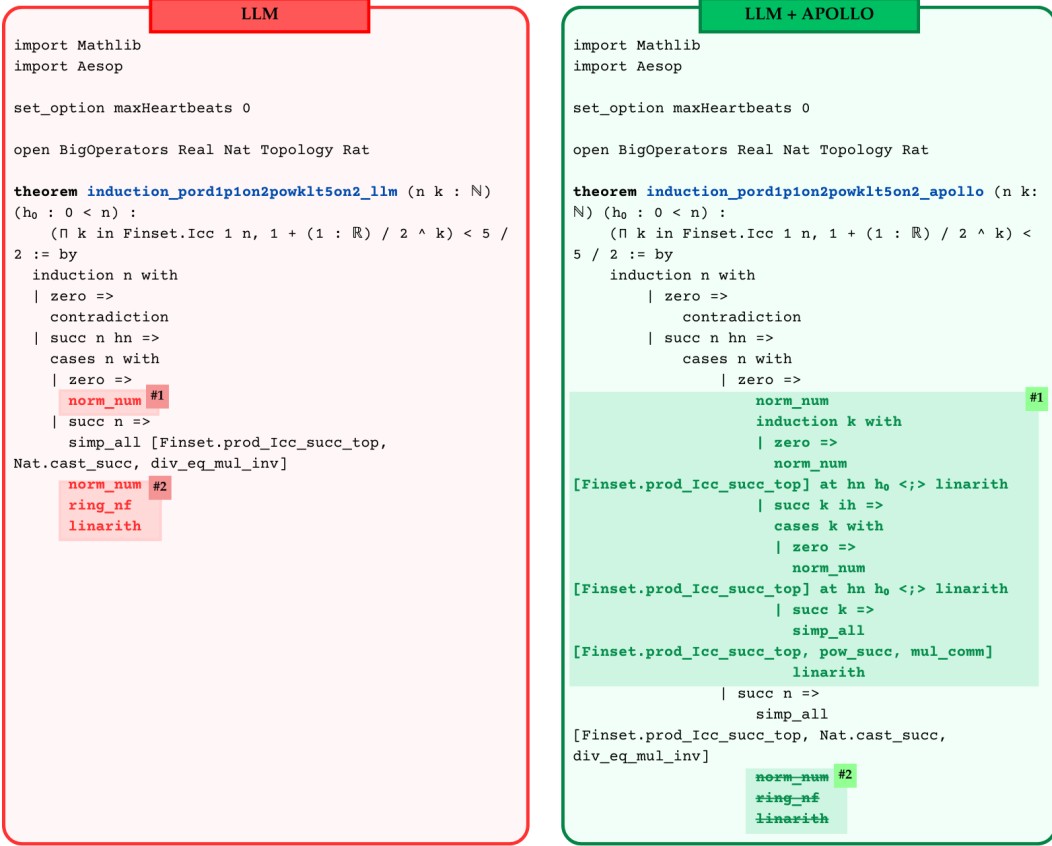

Figure 10: Proof attempt of `induction_pord1p1on2powklt5on2` produced by the base model (Goedel-Prover-SFT) versus after Apollo's targeted repair. The left shows the original end-to-end LLM output, which does not compile in Lean; the right shows the corrected, repaired proof assembled by Apollo that closes all of the goals.

```
LLM

import Mathlib
import Aesop

set_option maxHeartbeats 0

open BigOperators Real Nat Topology Rat

theorem mathd_algebra_293_llm (x : NNReal) :
    Real.sqrt (60 * x) * Real.sqrt (12 * x) * Real.sqrt
(63 * x) = 36 * x * Real.sqrt (35 * x) := by
  rw [← Real.sqrt_mul (by positivity), ←
Real.sqrt_mul (by positivity)]
  ring_nf
  rw [Real.sqrt_eq_iff_mul_self_eq]
  ring_nf
  nlinarith #1
```

```
LLM + APOLLO

import Mathlib
import Aesop

set_option maxHeartbeats 0

open BigOperators Real Nat Topology Rat

theorem mathd_algebra_293_apollo (x : NNReal) :
    Real.sqrt (60 * x) * Real.sqrt (12 * x) *
Real.sqrt (63 * x) = 36 * x * Real.sqrt (35 * x) :=
by
  rw [← Real.sqrt_mul (by positivity), ←
Real.sqrt_mul (by positivity)]
  ring_nf
  rw [Real.sqrt_eq_iff_mul_self_eq]
  ring_nf
                                              #1
  norm_num
  have h : (√(x : ℝ) * √35)^2 = (x : ℝ) * 35 := by
    rw [mul_pow]
    simp
  rw [h]
  ring
  simp_all only [ofNat_pos,
mul_nonneg_iff_of_pos_right, NNReal.zero_le_coe,
pow_nonneg]
  have hx : x ≥ 0 := by
      exact _root_.zero_le x
  have h1 : √((↑x : ℝ) * √(35 : ℝ)) *
(36 : ℝ) ≥ 0 := by
      simp_all only [ge_iff_le, _root_.zero_le,
NNReal.zero_le_coe, sqrt_mul, ofNat_pos,
        mul_nonneg_iff_of_pos_right,
Real.sqrt_pos, Real.sqrt_nonneg]
  have A : 0 ≤ (x : ℝ) := NNReal.coe_nonneg x
  have B : 0 ≤ √((x : ℝ) * 35) := sqrt_nonneg
((x : ℝ) * 35)
  have C : (0 : ℝ) ≤ 36 := by norm_num

  exact mul_nonneg (mul_nonneg A B) C
```

Figure 11: Proof attempt of `mathd_algebra_293` produced by the base model (Goedel-Prover-SFT) versus after Apollo's targeted repair. The left shows the original end-to-end LLM output, which does not compile in Lean; the right shows the corrected, repaired proof assembled by Apollo that closes all of the goals.

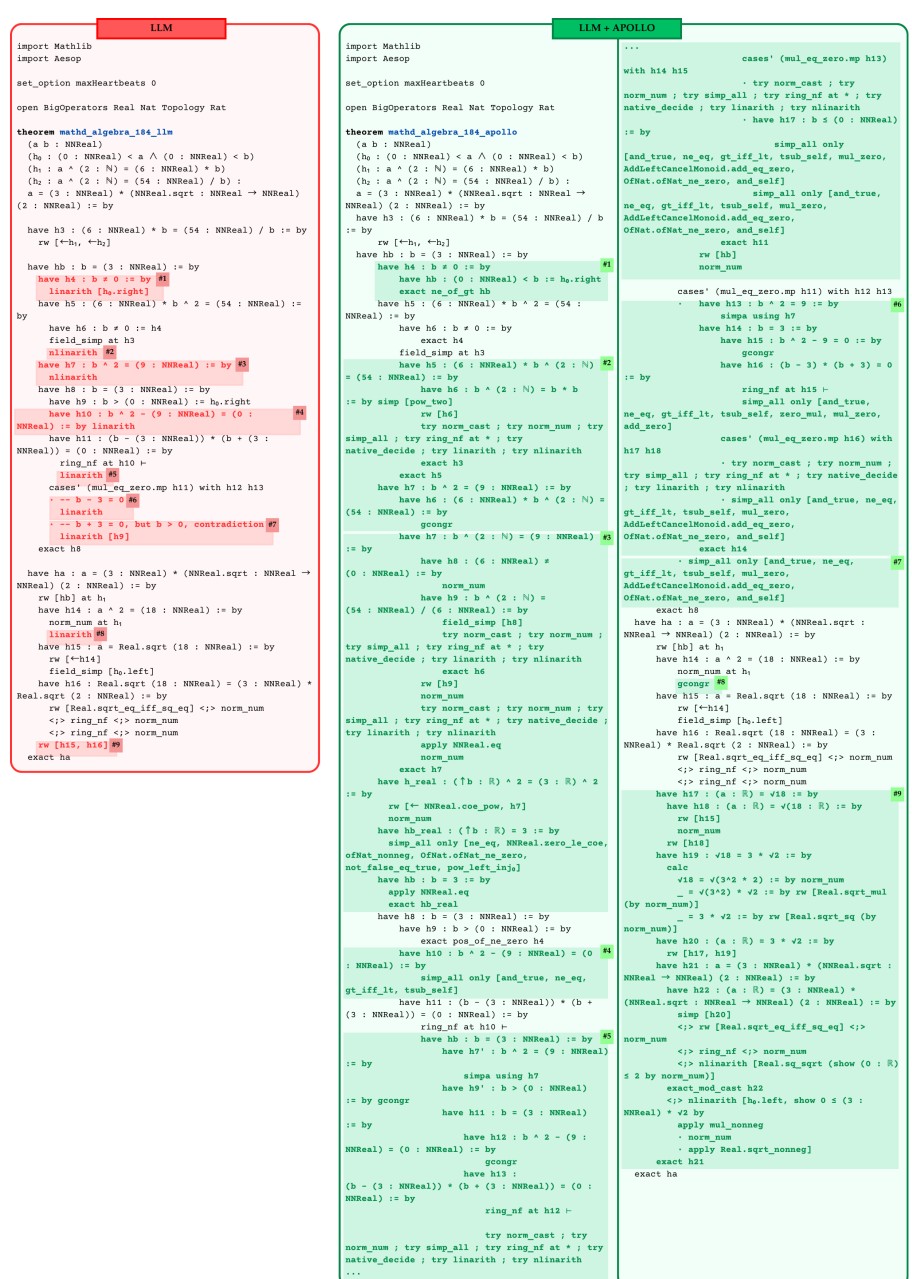

Figure 12: Proof attempt of `mathd_algebra_184` produced by the base model (Kimina-Prover-Preview-Distill-7B) versus after Apollo's targeted repair. The left shows the original end-to-end LLM output, which does not compile in Lean; the right shows the corrected, repaired proof assembled by Apollo that closes all of the goals.

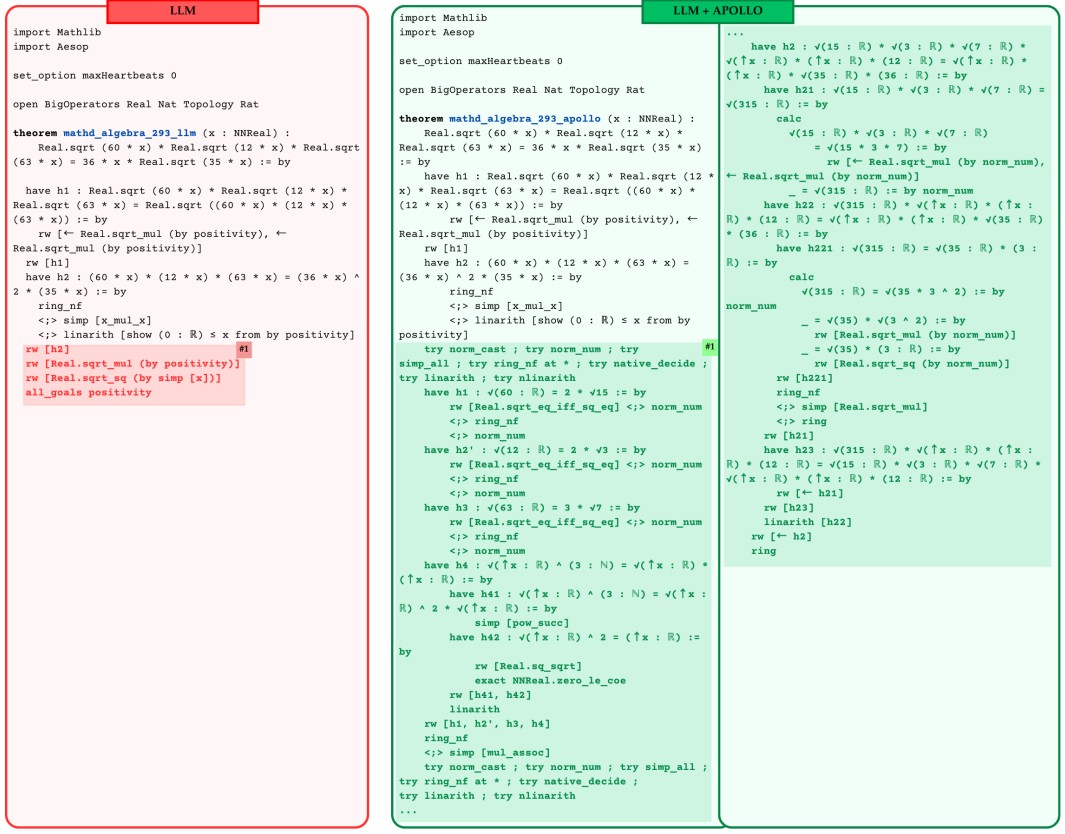

Figure 13: Proof attempt of `mathd_algebra_293` produced by the base model (Kimina-Prover-Preview-Distill-7B) versus after Apollo's targeted repair. The left shows the original end-to-end LLM output, which does not compile in Lean; the right shows the corrected, repaired proof assembled by Apollo that closes all of the goals.

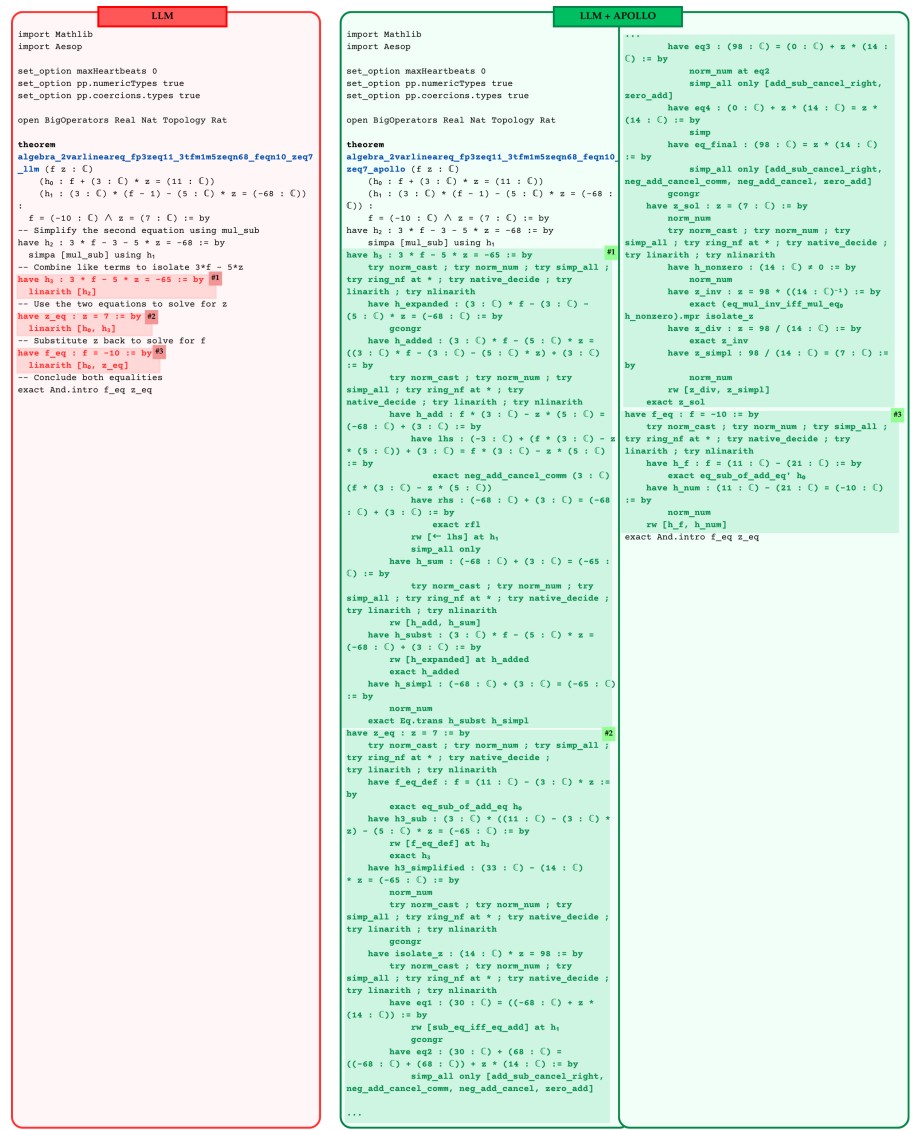

Figure 14: Proof attempt of `algebra_2varlineareq_fp3zeq11_3tfm1m5zeqn68_feqn10_zeq7` produced by the base model (o4-mini) versus after Apollo's targeted repair. The left shows the original end-to-end LLM output, which does not compile in Lean; the right shows the corrected, repaired proof assembled by Apollo that closes all of the goals.

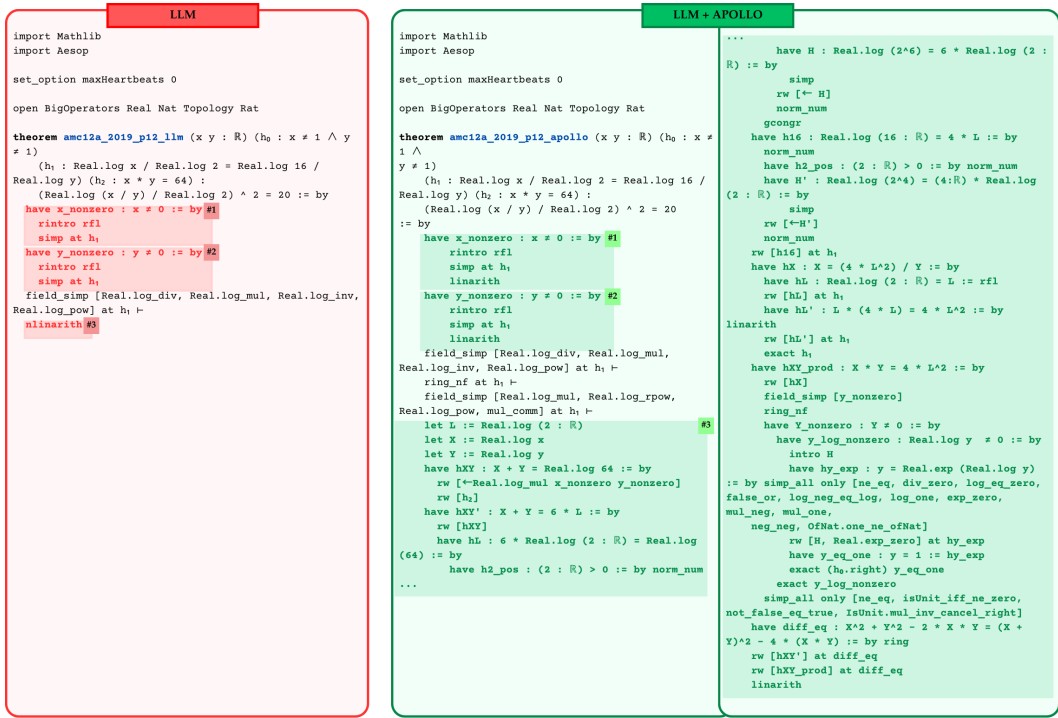

Figure 15: Proof attempt of `amc12a_2019_p12` produced by the base model (Goedel-Prover-SFT) versus after Apollo's targeted repair. The left shows the original end-to-end LLM output, which does not compile in Lean; the right shows the corrected, repaired proof assembled by Apollo that closes all of the goals.

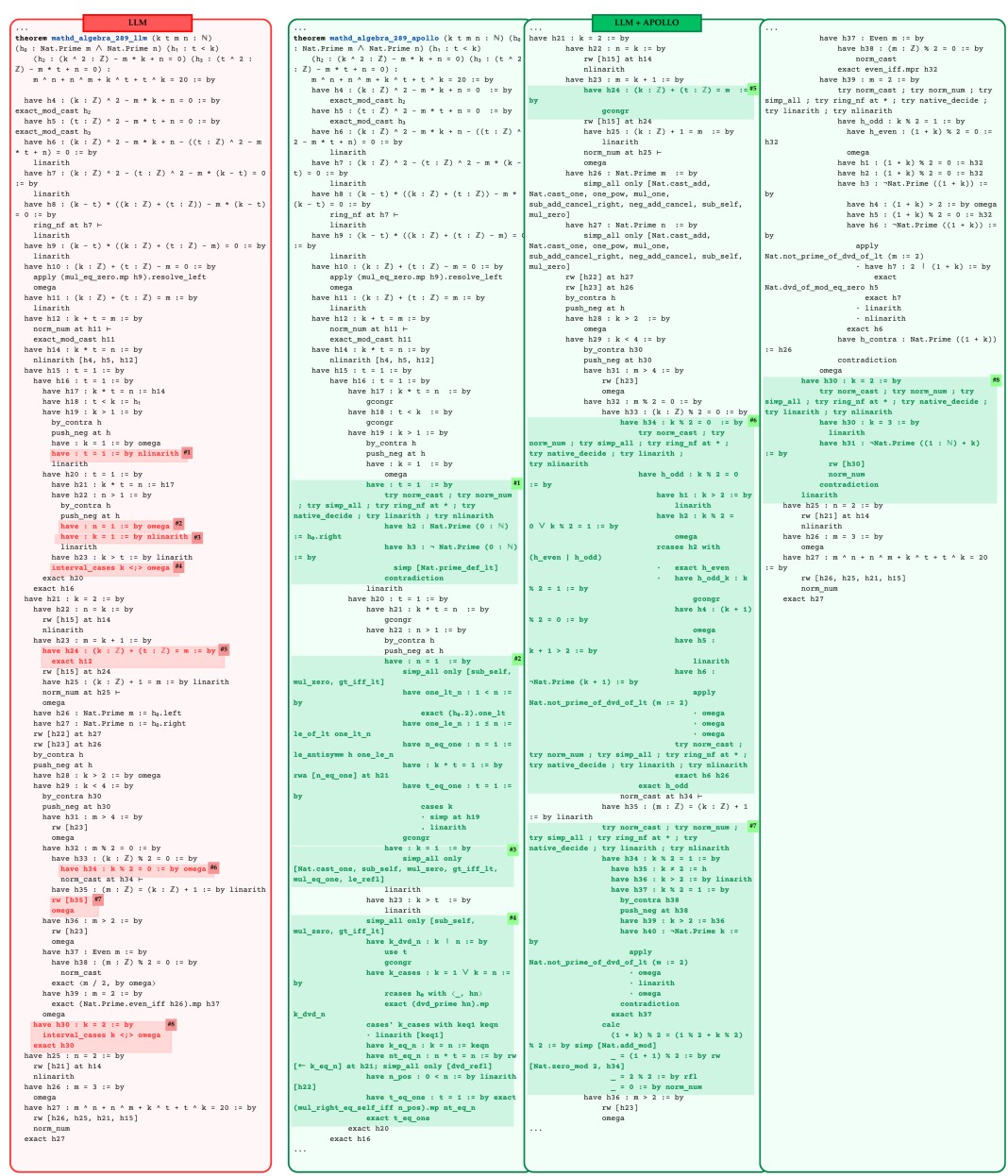

Figure 16: Proof attempt of `mathd_algebra_289` produced by the base model (Kimina-Prover-Preview-Distill-7B) versus after Apollo's targeted repair. The left shows the original end-to-end LLM output, which does not compile in Lean; the right shows the corrected, repaired proof assembled by Apollo that closes all of the goals. Import headers are the same as other presented examples, for visibility they were omitted in this diagram.

