# OpenReview forum: "APOLLO: Automated LLM and Lean Collaboration for Advanced Formal Reasoning"
_NeurIPS.cc/2025/Conference — NeurIPS 2025 poster_

### Official Review · Reviewer_gYTQ · 2025-06-30

**Clarity:** 3
**Significance:** 3
**Originality:** 2
**Rating:** 5
**Confidence:** 4

**Summary:**

The paper proposes APOLLO, a modular, agentic pipeline that incorporates signals from lean compiler with LLM prover to achieve better performance in mathematical theorem proving. The proposed system includes a set of agents to analyze the proofs, fix the syntax errors, identify the mistakes, decompose failing lemmas and utilize automated solvers to boost the success rate of the whole proofs generated by an LLM prover. With APOLLO, the authors achieve 75% accuracy on miniF2F test set and at the same time improve the LLM prover’s sample complexity. They also demonstrate its wide applicability to general-purpose reasoning models such as o3 and o4.

**Questions:**

I will update the score if the authors address some/all the weakness points I listed in the last section. Additionally, I have the following questions.

1.	In Fig. 3, the curve for APOLLO is cutoff at a relatively low sample budget. It’s important to understand how the accuracy scales with larger sample budge. Does APOLLO continue to prove more theorems or does it plateau and thus cannot further improve much more than the base model.

2.	In Fig. 4, is the data included in the figure conditioned on the proof being correct? I think it only makes sense if the authors show the results on correct proofs only. Making the wrong proofs longer is not interesting or important.

3.	How does the hyperparameter r transfer to sample budget? I suggest including a plot showing the relationship between r and sample budget.

4.	Is it possible to do pass@N on top of APOLLO? Is there a tradeoff between increasing r and increasing passes?

**Ethical Concerns:**

["NO or VERY MINOR ethics concerns only"]

**Final Justification:**

The paper was unclear about how they compare against the baseline. The authors has clarified this main concern. Given the fact that the authors have addressed most of my concerns, I increase my score to 5 and suggest for acceptance.

**Limitations:**

The authors have included a limitations section in the appendix. I agree that not considering step-level LLM prover is a main limitation of the work.

**Quality:**

3

**Strengths And Weaknesses:**

The paper is well written, and the authors have shown a good amount of improvement with their proposed APOLLO. However, there is still space to improve:

1.	The authors claim that they achieve state-of-the-art accuracy, but Deepseek-prover-V2-7B has an accuracy of 82%. I suggest the authors include this reference.

2.	I am not sure if the comparison of sampling budget is fair. Based on my understanding, the mean number of samples for APOLLO is computed as the number of function calls until it successfully proves the theorem or reaches a maximum recursion depth. Then to make fair comparison, I think for pass@k, the authors should also report the mean of number of samples until it FIRST samples a correct proof. I believe this number will be lower than the sample budget reported in the table.

3.	The proposed method APOLLO acts like a search tree. As a result, it may be important to include comparisons with tree-search methods such as Monte Carlo Tree Search and check how much APOLLO improves over those tree-search methods.

---

> ### Author Rebuttal · Authors · 2025-07-31
>
> We would like to thank Reviewer gYTQ for the constructive feedback and suggestions on our work. The following is our response to the reviewer’s comments and questions.
>
> ### **1. Absence of Deepseek-Prover-V2 from Related Works**
> Deepseek-Prover-V2 got released on the edge of the submission deadline and we could not properly test the model with Apollo. To remedy this, we performed experiments on the current state-of-the-art medium-sized prover: Goedel-V2-8B [1]. Our experiments show that Goedel-V2+Apollo solves four more problems in miniF2F-test, which constitutes a _1.6% gain in accuracy_.
>
> We will add the discussion of newer whole-proof generation models in the revised manuscript.
>
> ### **2.  Comparing true sampled budgets of Apollo and base provers**
> We agree with the reviewer that comparing an upper bound to the fixed passK parameter is not entirely fair. To remedy this, we consider two settings: observed upper bound and average number of sampled proofs across the entirety of miniF2F-test, including the failed proofs. For Apollo, we consider only theorems that were not proved by the base LLM, and for the base LLM we consider all attempted problems. The results are below:
>
> Table 1: Average and upper bound sampling budgets for base LLMs and Apollo aided proofs
> | model               | upper bound | average budget | accuracy |
> |---------------------|-------------|----------------|----------|
> | goedel-sft          |       25600 |           9730 |    64.7% |
> | kimina-7b           |        1024 |            373 |    70.8% |
> | goedel-sft + Apollo |        1100 |            618 |    65.6% |
> | kimina-7b + Apollo  |         888 |            367 |    75.0% |
>
>
> We observe that the average number of sampled proofs increases for Apollo when failure cases are included. For Goedel-SFT, gains remain substantial, as the base LLM attempts each problem many times up to 25,600. For Kimina-Prover, the average budgets with and without Apollo are similar, yet Apollo achieves higher accuracy with a lower maximum number of sampled proofs. Additionally, miniF2F contains a sizable number of easy problems that LLMs typically solve within 1–8 attempts, skewing base-LLM results toward lower counts. We omit these easier theorems from the Apollo evaluation, since Apollo did not participate in their proofs. Even under these conditions, Apollo matches or reduces sample complexity compared to the base LLMs while delivering accuracy improvements across the board.
>
> ### **3. Comparing Apollo to MCTS-based sampler**
> We would like to clarify that the Apollo framework differs from traditional tree-search methods. Our approach decomposes a theorem into independent sub-proofs, identifies the failing ones, and proves them recursively. Because each sub-goal is independent, these repairs can be performed in parallel. Unlike tree search, Apollo does not conduct an explicit search during inference; instead, it iteratively patches the initial proof attempt until it reaches a correct solution.
>
> To compare Apollo with a different MCTS-based sampling strategy, we have run Kimina-Prover-Distill-7B with the RMaxTS search method proposed by [2]. The results are below:
>
> Table 2: Comparing Apollo to RMaxTS sampling strategy
>
> | method        | budget (max sample count) | budget (tokens) | accuracy |
> |---------------|---------------------------|-----------------|----------|
> | kimina-7b        |             32            |       140K      |   63.1%  |
> | kimina-7b        |            1024           |       4.5M      |   70.8%  |
> | kimina-7b+RMaxTS |          4 x 128          |       2.0M      |   64.3%  |
> | kimina-7b+Apollo |            589            |       2.2M      |   75.0%  |
>
> With similar token budgets, Apollo significantly outperforms RMaxTS strategy and vanilla whole proof generation method. Although Apollo’s worst-case budget slightly exceeds RMaxTS, its average budget is lower. Apollo generated 307 proofs and 1.3M tokens on average, which is substantially lower than the RMaxTS strategy.
>
> ### **4. Accuracy scale with large recursion depth $r$**
> We decided to stop sampling once the performance curve plateaued and examined the outputs. Because Apollo’s effectiveness relies heavily on the underlying base model, it diminishes when the LLM has little idea on how to proceed. Our analysis shows that many failures arise from models re-proving given steps or hallucinating. Increasing recursion depth yielded no accuracy gains. We believe this plateau reflects the prover’s inherent quality, its problem-solving ability and capacity to generate high-quality proof sketches. At the time of this work, proof sketching existed only in external tools like DSP [3], not within provers themselves. We expect this to change soon, as new theorem provers begin to include high-level proof sketches and informal reasoning in their chain-of-thought outputs.
>
> ### **5. Subset of reported proofs in Figure 4**
> Yes, the figure only visualizes the proof length of valid proofs. We will make this point clear in the revised manuscript.
>
> ### **6. Relationship between $r$, sample budget and token count**
> Below are tables of the relationship between sample budget, recursion depth and token count for two theorem provers: Kimina-Prover-Preview-Distill-7B and Goedel.
>
> Table 3: Relationship between $r$, sampling budget and token count on Kimina-Prover-Preview-Distill-7B.
> | rec depth     |    0 |    2 |    4 |    6 |
> |---------------|-----:|-----:|-----:|-----:|
> | sample budget |   32 |   96 |  191 |  307 |
> | token count   | 140K | 406K | 816K | 1.3M |
>
>
> Table 4: Relationship between $r$, sampling budget and token count on Goedel-SFT.
> | rec depth     |     0 |     2 |     4 |    6 |
> |---------------|------:|------:|------:|-----:|
> | sample budget |    32 |    80 |   151 |  362 |
> | token count   | 16.1K | 38.3K | 74.6K | 179K |
>
> We observe that the sample budget and token count depend linearly on the recursion depth $r$. We will convert these tables into plots and include them in the revised manuscript.
>
> ### **7. Applying @N on top of Apollo**
> Yes, it is possible to apply @N at each iteration of the framework. Increasing @N results in more sampled proofs per sub-goal, giving the LLM additional opportunities to generate a valid proof. Conversely, increasing $r$ leads to deeper sub-goal decomposition and greater simplification of each sub-goal. A higher $r$ parameter allows Apollo to explore each sub-goal more thoroughly and increases the likelihood that it will be further decomposed.
>
> Throughout our experiments with @N fixed at each recursion iteration, we consistently improved the baseline accuracies of all theorem provers while reducing computational costs. Goedel-SFT’s accuracy increased by 0.9%, and its sampling budget decreased from 25,600 to 362. Kimina-Prover-Preview-Distill-7B saw a 4.2% improvement while using only one-third of the sampling budget. General-purpose models jumped from a few percent to over 40% accuracy. Our experiment also shows that the performance of even the latest theorem provers, such as Goedel-V2, can be further enhanced. Increasing the @N parameter could yield additional gains and serve as an extra hyperparameter in the Apollo framework.
>
> ---
>
> Thank you again for your thorough feedback. Your comments have helped clarify our paper, and the revised version will offer deeper insights into agentic Apollo repair. In light of these improvements, we hope you might consider increasing your score. If you have any further questions, please don’t hesitate to let us know.
>
> ---
>
> [1] Yong Lin, et al. (2025). Goedel-Prover-V2: The Strongest Open-Source Theorem Prover to Date.
>
> [2] Huajian Xin, et al. (2025). DeepSeek-Prover-V1.5: Harnessing Proof Assistant Feedback for Reinforcement Learning and Monte-Carlo Tree Search. In The Thirteenth International Conference on Learning Representations.
>
> [3] Albert Qiaochu Jiang, et al. (2023). Draft, Sketch, and Prove: Guiding Formal Theorem Provers with Informal Proofs. In The Eleventh International Conference on Learning Representations.

---

> > ### Comment · Reviewer_gYTQ · 2025-08-02
> >
> > I thank the authors for addressing most of my concerns and questions. I have a few additional questions:
> >
> > The authors acknowledge that comparing an upper bound to a fixed pass@K parameter may not be entirely fair. However, it remains unclear how exactly the newly computed average budget is derived. Specifically, the authors state: "For Apollo, we consider only theorems that were not proved by the base LLM, and for the base LLM we consider all attempted problems." Could the authors further clarify what is meant by "only considering theorems that were not proved by the base LLM"? Moreover, how is the sample budget computed for theorems that were proved by the base LLM when evaluating Apollo? When comparing the methods, I believe it is important to make sure the data are kept the same.
> >
> > Additionally, I am curious whether the proposed method could be effectively integrated into a step-level prover. Could the authors comment on this potential extension?

---

> > > ### Author Response · Authors · 2025-08-04
> > >
> > > We thank Reviewer gYTQ for the thoughtful feedback on our response. Regarding the points raised in the feedback:
> > >
> > > > Could the authors further clarify what is meant by "only considering theorems that were not proved by the base LLM"? Moreover, how is the sample budget computed for theorems that were proved by the base LLM when evaluating Apollo?
> > >
> > > We thank the reviewer for raising a critical question regarding our budget computation methodology. In short, all the results that we have reported so far are on the subset of difficult problems in miniF2F that LLMs were unable to write a correct proof for given a certain number of attempts. To further clarify our findings, we report Apollo’s budget, both sample count and token usage, in two scenarios:
> > >
> > > 1. **Entire miniF2F-test dataset**: We measure the full pipeline’s budget regardless of Apollo’s intervention. Even if the base LLM solves a problem in one attempt, that attempt counts toward “base LLM + Apollo.” The results are reported in Table 1A.
> > >
> > > 2. **Apollo-assisted proofs only**: We exclude problems the base LLM solved unaided and focus solely on proofs where Apollo was invoked, that is, where the base LLM initially failed. This scenario is reported in Table 2A and it mirrors the setting of Table 1 from our previous rebuttal response.
> > >
> > > Tables 1A and 2A report the budgets for each scenario. To further expand the findings, we also provide token budgets along with sample budgets.
> > >
> > > Table 1A: Average and maximum budget of “base LLM + Apollo” on the entire miniF2F-test
> > > | model               | max budget | average budget | max token count | average token count |
> > > |---------------------|------------|----------------|-----------------|---------------------|
> > > | goedel-sft + Apollo |       1100 |            335 |            548K |                167K |
> > > | kimina-7b + Apollo  |        888 |            189 |            3.8M |                818K |
> > >
> > > Table 2A: Average and maximum budget of “base LLM + Apollo” on the Apollo invoked subset of miniF2F-test
> > > | model               | max budget | average budget | max token count | average token count |
> > > |---------------------|------------|----------------|-----------------|---------------------|
> > > | goedel-sft + Apollo |       1100 |            618 |            548K |                307K |
> > > | kimina-7b + Apollo  |        888 |            367 |            3.8M |                1.6M |
> > >
> > > We observe that in scenario (1), the average token and sample budgets are substantially lower than in scenario (2) because it includes the subset of problems in miniF2F that are easier and the base LLM can prove unaided. Table 1A which we will add to the manuscript, provides insights on how Apollo affects the total budget and accuracy of ATP on the entire miniF2F dataset. Table 2A which we have already reported in the rebuttal, highlights Apollo’s contribution on the more challenging subset of the miniF2F dataset, the problems that Apollo has engaged in writing their proofs and the LLM by itself was unable to prove. We believe reporting scenario (2) is more informative (our original reporting), but, per your suggestion, we will report scenario (1) so the reader will have all the information. We hope this clears up any confusion around the reported sample budgets.
> > >
> > > > I am curious whether the proposed method could be effectively integrated into a step-level prover. Could the authors comment on this potential extension?
> > >
> > > That’s a great question, Apollo can already be used in conjunction with step provers. The best step prover in the literature, BFS-prover, has not released its tree search code and it’s not trivial to reproduce their results, so we chose to experiment with the best whole proof generation LLMs such Kimina Prover, and most recently, with Goedel Prover v2. However, one can switch the LLM in our code base to any functioning step prover, and use Apollo to improve the accuracy of such step prover.
> > >
> > > The community evaluates the accuracy of step provers the same way as the accuracy of whole proof generation LLMs. In other words, step provers also write complete proofs and their accuracy is measured by evaluating whether that whole proof is correct or not, albeit their mechanism of proof generation is different. Therefore, in any instance where a whole proof generated by a step prover has flaws, Apollo can work on that proof and try to fix the errors in the proof. Apollo is applicable to step provers as much as it is applicable to whole proof generation models.
> > >
> > > A possible direction for future research is to use Apollo to intervene during the proof generation process of step provers, and try to fix any errors before the whole proof is written. Such early engagement/intervention during the proof generation process is likely to further reduce the computational cost (token budget).

---

> > > > ### Comment · Reviewer_gYTQ · 2025-08-04
> > > >
> > > > I thank the authors for further clarification. I believe reporting numbers on the entire test set is more fair. I have increased my score to 5 as the paper has been improved.

---

### Official Review · Reviewer_KutC · 2025-07-02

**Clarity:** 3
**Significance:** 3
**Originality:** 3
**Rating:** 4
**Confidence:** 4

**Summary:**

The paper proposes an inference-time workflow for theorem proving with large language models (LLMs) in Lean, which comprises a series of specialized tools that fix up common syntax errors in LLM-generated code, remove parts with errors, apply a suite of Lean's standard proof automation tools and otherwise recurse by extracting the remaining proof gaps as standalone proving problems for the LLM.

Compared to the baseline of $k$ independent samples from the prover LLM, this post-processing and recursive repair significantly reduces cost / outperforms the baseline on the standard MiniF2F benchmark when comparing at the same amount of LLM calls. The method can be applied to any prover model and is both used with Lean-finetuned models like Goedel-Prover and general-purpose reasoning models such as o3-mini.

**Questions:**

see weaknesses 1-3: comparison to agentic setup? CPU times? ablations?

**Ethical Concerns:**

["NO or VERY MINOR ethics concerns only"]

**Final Justification:**

Following the discussion, I increase my Significance rating (there is no careful analysis on agentic harnesses for Lean) and decrease my  Quality rating (important considerations and their interactions are under-explored, the comparison should include iterative instead of recursive repair, ablate LLM repair and rule-based repair separately and report early stopping numbers for the pass@k baseline just like for the repair methods). I still marginally recommend the paper given that it opens up these questions and provides valuable insights from the available datapoints already.

**Limitations:**

yes

**Quality:**

2

**Strengths And Weaknesses:**

**Strengths:**
1. The post-processing and recursive repair appear extremely efficient at catching and fixing minor mistakes in the code, producing strong evaluation numbers on a variety of models.
2. The different post-processing stages are clearly described and the paper is easy to understand.

**Weaknesses:**
1. The method is not compared against other inference-time methods such as putting the model into a multi-turn agentic loop where it reads in error messages and can decide freely how to solve the issue (by replacing a block, using automation tools, reworking a line etc). I understand that this is in part due to the fact that whole-proof generation models such as Goedel-Prover are not able to follow such instructions, but o3-mini and o4-mini certainly could.
2. The comparison based on LLM calls is fair as long as GPU compute is the primary limiting factor. Given that in Lean proving setups, CPU execution is an increasingly important bottleneck, it would be important to see Lean execution runtimes, too, for getting the full picture.
3. There are no ablations on the different parts of the post-processing pipeline. At least, it would be useful to see average numbers on how often which tool triggers with which model. I would be surprised, for instance, if the Syntax Refiner often triggers for the Lean4-specific models. Moreover: How many gaps can be solved with the Lean automation? How many recursive calls are being issued on average? (Table 1 lists the average total number of calls per problem but doesn't state from how many independent attempts these stem.)
4. The method proposes ad-hoc fixes to rather superficial problems of current LLMs for Lean (confusion between Lean 3 and Lean 4 among general-purpose models; incomplete knowledge of all automation tactics), it is conceivable that the method will not be relevant for a very long time.

---

> ### Author Rebuttal · Authors · 2025-07-31
>
> We would like to thank Reviewer KutC for the constructive feedback and suggestions on our work. The following is our response to the reviewer’s comments and questions.
>
> ### **1. Apollo against iterative feedback repair**
>
> Indeed, iterative repair frameworks have proven successful and are widely adopted in the code-repair community. To evaluate how Apollo compares, we conducted a simple experiment using o3-mini as the base model. Over up to ten iterations, the model takes the incorrect proof along with feedback from the Lean compiler as an input and produces a corrected proof. Our findings are detailed below:
>
> Table 1: Comparing feedback repair to Apollo repair with o3-mini as the base model
> | method                    | budget | accuracy |
> |---------------------------|--------|----------|
> | o3-mini                   |      1 |    3.3% |
> | o3-mini                   |     32 |   24.6% |
> | o3-mini + feedback repair |      5 |   17.2% |
> | o3-mini + feedback repair |     10 |   25.4% |
> | o3-mini + Apollo          |     ~8 |   40.2% |
>
> Feedback prompt schema:
> ```
> This is an incorrect proof:
> [model’s last proof]
> Compilation errors are as follows:
> [Lean’s error messages]
> Based on this feedback, produce a correct raw Lean code for the following problem:
> [header]
> [informal prefix]
> [formal statement]
> ```
> To the best of our knowledge, there is no reliable method to obtain reasoning tokens via API for o-series models, so we report the results based on average pass parameters. We observe that Apollo outperforms this method with a lower average budget.
>
> ### **2. CPU and GPU bottlenecks in theorem proving**
>
> Indeed, our focus on cutting the generated sample complexity is motivated by our observations of resource allocation. In our setup, CPU is not a limiting factor. Even with high parallelization, the Lean kernel is efficient and proof verification is fast. Moreover, to avoid verifying proofs with too many recursive calls and inefficient structures, we set a 5 minute limit on compilation. From what we have seen, compilers rarely get close to exhausting the time limit and on average proof verification takes between 6-200 seconds depending on its complexity.
>
> By contrast, GPU resources are a substantial bottleneck in our environment. Running Kimina-Prover-Preview-Distill-7B on a single A5000 GPU with a 16,384-token context window takes 700 to 2,000 seconds per problem, over 11.5 minutes for @32 sampling. Increasing sampling cost makes this bottleneck worse. Even with eight GPUs and parallel generation, throughput remains slow and difficult to scale. Therefore, we focused not only on improving theorem-proving accuracy but also on minimizing the number of LLM invocations.
>
> We also note that Lean-based servers are becoming more efficient. Kimina-server enables parallelization across CPU cores and accelerates proof verification. Going forward, we believe that increasing sampling costs will impose greater overhead on GPUs than on CPUs.
>
> We will include this discussion in the revised manuscript.
>
> ### **3. Additional information on connection between sampling costs and recursive calls**
>
> To further clarify the connection between recursive calls and sampling costs, we present the following tables that reports the recursive calls, sampling costs and token usage for Goedel-Prover-SFT (goedel-sft) and Kimina-Prover-Preview-Distill-7B (kimina-7b). These results are an extension of Table 1 (main text) and they show connection between sample budget and recursion depth.
>
> Table 1: Relationship between $r$, sampling budget and token count on Kimina-Prover-Preview-Distill-7B.
> | rec depth     |    0 |    2 |    4 |    6 |
> |---------------|-----:|-----:|-----:|-----:|
> | sample budget |   32 |   96 |  191 |  307 |
> | token count   | 140K | 406K | 816K | 1.3M |
>
>
> Table 2: Relationship between $r$, sampling budget and token count on Goedel-SFT.
> | rec depth     |     0 |     2 |     4 |    6 |
> |---------------|------:|------:|------:|-----:|
> | sample budget |    32 |    80 |   151 |  362 |
> | token count   | 16.1K | 38.3K | 74.6K | 179K |
>
> ### **4. Ablation studies**
>
> We thank the reviewer for the comment on ablating the different parts of the pipeline. We performed ablation using two baselines: o4-mini and Kimina-Prover-Preview-Distill-7B. Below is the ablation study and each module’s contribution. The starting point is @32 for Kimina and @1 for o4-mini. Recursive repair is up to $r=6$ for Kimina and $r=4$ for o4-mini, while recursive repair for “base+SR+Auto Solver” case is $r=1$, since without invoking LLM Solver the sub-goals cannot be expanded and decomposed further. Results are shown in Table 3.
>
> Table 3: Ablation study of different parts of Apollo on o4-mini (general purpose model) and Kimina-Prover-Preview-Distill-7B (theorem prover model)
> | module                   | o4-mini | kimina-7b |
> |--------------------------|---------|-----------|
> | base                     |    7.0% |     63.1% |
> | base+Syntax Refiner (SR) |    7.4% |     63.1% |
> | base+Auto Solver         |    7.0% |     63.5% |
> | base+SR+Auto Solver      |   20.5% |     63.5% |
> | base+LLM Solver          |    8.2% |     69.3% |
> | base+SR+LLM Solver       |   18.9% |     69.3% |
> | base+Apollo              |   46.7% |     75.0% |
>
> For general-purpose models like o4-mini, the Auto Solver delivers substantial benefits: these models rarely invoke built-in solvers and instead attempt to prove simpler goals manually, often resulting in hallucinations and redundancy. Applying the LLM Solver alone yields lower performance. Syntax Repair by itself fixes only one problem, but it’s critical for enabling other modules, without it, o4-mini shows little improvement. Although we expect the Syntax Refiner to become obsolete soon, we include it in the pipeline to support experiments with general-purpose models.
>
> By contrast, Kimina-Prover follows a different pattern. The Syntax Refiner has no impact, this model is fine-tuned to produce only syntactically valid proofs. Auto Solver alone aids only one theorem, while repeated LLM Solver invocations yield a 6% improvement. We see a substantial performance boost when both modules are combined. Upon inspecting Apollo’s outputs, we find that the model often makes minor errors while preserving the correct proof skeleton, and many sub-goals can then be discharged with built-in tactics. Thus, the interaction of both modules produces significant accuracy gains.
>
> To further investigate each module, we conducted a simple test to see how many times each module was called during an inference.
>
> Table 4: Number of Apollo module triggers across different models
> | model      | Syntax Refiner | Auto Solver | LLM Solver |
> |------------|----------------|-------------|------------|
> | goedel-sft |           0.0% |      100.0% |     100.0% |
> | kimina-7b  |           0.0% |      100.0% |      95.2% |
> | o3-mini    |         100.0% |      100.0% |      70.0% |
> | o4-mini    |          94.7% |      100.0% |      64.9% |
>
> Across all experiments, the Auto Solver is always invoked to apply Lean solvers to each sub-goal; if it fails, it leaves the goal open and passes it to the LLM Solver. Conversely, the LLM Solver is sometimes bypassed when the Auto Solver discharges a goal without calling external provers. For Goedel-SFT and Kimina-Prover, the Syntax Refiner is never invoked. We evaluate only those problems proved with Apollo’s assistance. Note that the Sorrifier and Proof Assembler are always invoked, they decompose proof code and reassemble it, so they are omitted from Table 4.
>
> We will include these results in the revised manuscript to emphasize the significance of each part of the pipeline and how they affect the overall performance of the framework.
>
> ### **5. Relevance of recursive repair**
>
> We agree that the relevance of Syntax Refiner is short-lived and it depends on how long it will take for general-purpose models to see enough Lean4 data. However, other parts of the pipeline are meant to be modular and upgradeable. We are not tied to any specific LLM or Lean version. Going into the future, Apollo will get better with the introduction of new LLMs and built-in solvers in Lean.
>
> Moreover, the current research trajectory indicates that leveraging the Lean compiler is a powerful idea already gaining traction in developers of models like Kimina-Prover-72B [1] and Goedel-V2 [2]. Both of these models include “error correction” modules that use REPL feedback to fix proof mistakes and were released two months after the proposal of Apollo. We expect researchers will increasingly integrate Lean feedback into LLMs to boost performance and reduce hallucinations. Apollo represents a step toward agents that not only generate proofs with LLMs but also incorporate external tools and resources, strengthening performance while reducing sole reliance on ATPs.
>
> To further demonstrate the relevance of our approach, we conducted an Apollo repair on the state-of-the-art prover: Goedel-V2. Our preliminary results indicate that Apollo aided in solving four additional problems in the miniF2F-test, resulting in a _1.6% performance gain_. This model was released much later than our proposed method, and our results show that we can further improve the accuracy of existing provers, indicating the longevity of the proposed framework.
>
> ---
>
> Thank you once more for your insightful feedback. We’ve carried out additional experiments and incorporated your suggestions to strengthen our work. Given these enhancements, we kindly ask you to consider revising your score. Please let us know if you have any further questions or concerns.
>
> ---
>
> [1] Haiming Wang, et al. (2025). Kimina-Prover: Applying Test-time RL Search on Large Formal Reasoning Models.
>
> [2] Yong Lin, et al. (2025). Goedel-Prover-V2: The Strongest Open-Source Theorem Prover to Date.

---

> > ### Comment · Reviewer_KutC · 2025-08-06
> >
> > Thank you for the detailed answers and analyses.
> >
> > Based on sections 1 and 4 of the rebuttal, **iterative (agentic) repair may be a viable alternative to recursive repair**, provided that it is coupled with the rule-based repair procedures (syntax refinement + auto solver). (Just crudely estimating that if with rule-based repair, pass@1 is already 20% and 10 turns of iterative repair correspond to pass@32, it is conceivable that the resulting score would also be in the 40-45% ballpark like for recursive repair).
> >
> > This does not diminish the relevance of this work, but **calls for a careful analysis** of which component is necessary in which context (recursive vs iterative LLM repair, rule-based repair tools yes vs no, general purpose model vs Lean-specific model). The paper would greatly benefit if it was written from such an objective "analyze what works" perspective, as opposed to "here is our method, and it works".
> >
> > The paper should be updated with ablations such as 1, 3, 4 in this rebuttal and their interaction as explained above. As a side note, ablation 3 should report success rates for the auto solver, not whether it was called.
> >
> > Another **important objection** raised by Reviewer gYTQ calls for reporting early stopping numbers for the standard pass@k infererence too. For instance, in the worst case where 57.6% of samples need exactly 32 attempts with Goedel-Prover-SFT and 1.6% of samples need 64, and all others fail after 64, the early stopping effective k would be $32*0.576 + 64 * (1-0.576) = 45.568$ with 59.2% success, a point that lies **above** the loglinear interpolation of the two closest Apollo points.
> >
> > The paper should include early stopping metrics (setting $k_{\text{effective}} = \frac{1}{p}$ for a problem with passrate $p$), and also include points in the interval between 64 and 3200 where the datapoints for the recursive repair method reside. Indeed, all these numbers can be computed solely from the existing pass@25600 evaluations, no additional experiments are required.

---

> > > ### Author Response · Authors · 2025-08-07
> > >
> > > We thank Reviewer KutC for the thoughtful feedback on our response.
> > >
> > > We will include the table comparing recursive repair with feedback-based repair. Since compiler-guided self-repair is a relatively recent development in the Lean ATP literature, it was not discussed in the submitted draft. We intend to expand this section and elaborate on the strengths and weaknesses of both iterative and recursive approaches, especially given that recent provers, such as Kimina-Prover-7B and Goedel-V2, support built-in iterative repair.
> > >
> > > Adding an early-stopping metric is a great suggestion, and we will include a discussion of when recursive repair halts to improve readers’ understanding of the Apollo framework. We appreciate the reviewer for raising this important point.
> > >
> > > Finally, we will expand the presented plots and tables and include all discussed ablation studies to analyze the contribution of each Apollo module and how they affect the prover’s problem-solving abilities.

---

### Official Review · Reviewer_hqN5 · 2025-07-02

**Clarity:** 3
**Significance:** 2
**Originality:** 2
**Rating:** 3
**Confidence:** 4

**Summary:**

The paper introduces Apollo, an automated modular pipeline that enhances large language model (LLM)-based theorem proving by integrating LLM outputs with feedback from the formal verification system (Lean4). Rather than naive resampling, Apollo iteratively identifies and repairs errors in LLM-generated proofs through a sequence of specialized modules. Specifically, the pipeline includes a Syntax Refiner that corrects superficial syntax issues; a module that replaces faulty sub-proofs with Lean's proof placeholders; an Auto Solver that applies built-in tactics to resolve sub-goals; and a recursive repair mechanism that re-invokes the LLM on remaining goals using Lean's feedback. Finally, the Proof Assembler re-integrates the repaired components and verifies the complete proof. The proposed method demonstrates strong empirical performance on the miniF2F benchmark across different base models.

**Questions:**

1) For a more rigorous evaluation, I suggest fixing the base model and comparing Apollo with alternative strategies under the same sampling budget. In particular, using the number of tokens as the budget unit rather than the number of sampled completions, would more accurately reflect computational cost. Additionally, evaluating the method on a broader set of benchmarks would help demonstrate the generalizability of the approach.
2) The core idea of identifying erroneous segments and selectively repairing them is conceptually straightforward. The contribution primarily lies in the agentic system design and modular workflow. Clarifying how Apollo differs from standard modular repair workflows or existing agentic frameworks would help assess its conceptual contribution.
3) The paper would benefit from a direct comparison with standard depth-first or best-first search strategies. In tree search, invalid branches are pruned early, potentially reducing token usage by avoiding downstream propagation of errors. In contrast, Apollo must retrospectively correct affected steps, which could be less efficient.
4) The descriptions of Apollo's modules suggest that they are LLM-based agents, but no prompt templates or examples are provided. Upon reviewing the source code, it appears that the modules are rule-based programs. If so, this raises concerns about scalability and generalizability: manually crafting rules for different failure modes is unlikely to scale across programming languages (e.g., from Lean to Prolog, even across different Lean versions), different LLMs (some correction rules are model-specific according to the code comments), or datasets. Clarifying this aspect and discussing its limitations would strengthen the paper.
5) Several components necessary for reproducibility are missing from the released source code, including a README, installation requirements, and datasets or download instructions.

**Ethical Concerns:**

["NO or VERY MINOR ethics concerns only"]

**Final Justification:**

The authors improved the quality of their work by adding additional explanation. Although not too impressive, the work shows some improvement relative to baselines. As such I had increased my initial rating by one.

**Limitations:**

yes

**Quality:**

2

**Strengths And Weaknesses:**

Strengths:
The paper is well-organized and easy to follow. The proposed method achieves strong performance on miniF2F benchmark with reduced sampling. The results analysis is thorough. The agentic pipeline is practical and aligns with human-like proof workflows.

Weaknesses:
Comparisons lack consistency in model and token budget; more datasets are needed. The method focuses on workflow design rather than algorithm or architecture-level innovation. No comparison with standard depth-first or best-first search methods. Modules rely on hand-crafted rules, limiting generalizability. Key artifacts (README, requirements, data) are missing from the code release.

---

> ### Author Rebuttal · Authors · 2025-07-31
>
> We would like to thank Reviewer hqN5 for the constructive feedback and suggestions on our work. The following is our response to the reviewer’s comments and questions.
>
> ### **1. Reporting token budget vs accuracy across the board.**
>
> > Comparisons lack consistency in model and token budget; more datasets are needed.
>
> As per your suggestion we are now presenting both generated sample counts together with generated tokens in all of our experiments across the board. In all the subsequent results, you will see comparison of tokens and accuracies for all of our findings except for o-series models. Across the board, for all LLMs, Apollo increases the accuracy while reducing the token count. We believe this addresses the first item under the weaknesses in your review. Thank you for this helpful suggestion which highlights the contribution of our work much clearer.
>
> Table 1: Performance of base Kimina-7B against LLM with Apollo on ProofNet dataset
> | method        | budget (highest sample count) | budget (tokens) | accuracy |
> |---------------|-------------------------------|-----------------|----------|
> | Kimina-7B        |                           128 | 559K            |   11.30% |
> | Kimina-7B+apollo |                            96 | 415K            |   18.30% |
>
> Table 2: Performance of base Kimina-7B against LLM with Apollo on PutnamBench dataset
> | method        | budget (highest sample count) | budget (tokens) | accuracy |
> |---------------|-------------------------------|-----------------|----------|
> | Kimina-7B        |                            32 | 180K            |        7 |
> | Kimina-7B        |                           192 | 1.1M            |       10 |
> | Kimina-7B+apollo |                           108 | 579K            |       11 |
>
> We observe that applying Apollo improves the accuracy across both benchmarks while reducing the number of generated tokens compared to the base LLM.
>
> ### **2. Limited generalizability**
>
> > Modules rely on hand-crafted rules, limiting generalizability.
>
> **We have now experimented with the Goedel Prover v2 8B model, the latest state-of-the-art theorem prover released in July and Apollo increases its accuracy by 1.6% while reducing the token budget.** Although this model was released two months after our method, our results demonstrate that Apollo can still enhance existing provers’ accuracy, highlighting the framework’s longevity and generalizability.
>
> It is also notable that two of the best models in the literature, Goedel Prover V2 and Kimina Prover have now added modules for error fixing using the list of errors generated by the Lean compiler. This is an indication of the trajectory of research in the community and the level of interest in adopting methods such as Apollo.
>
> Apollo is an agent that combines rules, LLMs, and built-in solvers to produce correct proofs. We acknowledge that some components, like the Syntax Refiner, which uses fixed rules to correct syntax mistakes, may become outdated as general-purpose models improve and stop making those mistakes. However, Apollo is fully modular and its modules can be added, altered, or removed, and possibly extended to new languages and use cases. LLMs can be swapped via a configuration file, and built-in solvers will continue to improve. Apollo does not depend on a specific Lean version (we tested v4.9.0 through v4.17.0), so upgrading individual components can yield better performance.
>
> To ensure generalizability, Apollo encodes code into an execution tree based on indentation and a few Lean-specific rules. Any language that uses indentation for code blocks could be supported with minimal overhead. The framework is designed to be modular and extensible: any module is replaceable and scalable.
>
>
> ### **3. Comparing Apollo to alternative sampling strategies.**
>
> > No comparison with standard depth-first or best-first search methods.
>
> Per your suggestion, we compare Apollo against RMaxTS [30], using the Kimina-Preview-Distill-7B model as the base. RMaxTS employs Monte Carlo Tree Search to explore the search space progressively, leveraging an “expand and truncate” strategy to adapt it for use with whole-proof-generation models.
>
> Table 3: Comparing Apollo to RMaxTS sampling strategy
> | method        | max sample count | tokens | accuracy |
> |---------------|---------------------------|-----------------|----------|
> | Kimina-7B        |             32            |       140K      |   63.1%  |
> | Kimina-7B        |            1024           |       4.5M      |   70.8%  |
> | Kimina-7B+RMaxTS |          4 x 128          |       2.0M      |   64.3%  |
> | Kimina-7B+Apollo |            589            |       2.2M      |   75.0%  |
>
> Per your request, we present Apollo’s maximum token expense to illustrate its efficiency in the worst-case scenario. As shown in Table 3, LLM with RMaxTS achieves 64.3%, while Apollo pushes the accuracy further to 75.0%, with 10% more tokens in the worst case generation scenario. Although Apollo’s worst-case budget slightly exceeds RMaxTS, its average budget is substantially lower. Apollo generated 307 proof attempts and 1.3M tokens on average.
>
> ### **4. Contribution of agentic Apollo framework**
>
> > Clarifying how Apollo differs from standard modular repair workflows or existing agentic frameworks would help assess its conceptual contribution.
>
> Although agentic workflow has an extensive and successful use in the general LLM literature, it has not been widely adopted in the ATP pipelines. We are now adding additional discussion and comparison with that literature outside the ATP to address this comment.
>
> Agentic LLM systems have long incorporated external tools, such as symbolic engines, APIs, search access, and memory modules, to improve performance and reduce hallucinations [1]. Recent multi-agent frameworks often focus on advanced inter-tool interactions [2] and complex agent-to-agent interactions [3], but the core insight remains: teaching LLMs to integrate outside resources is key. The outside source in our approach is the formal verification system: Lean. In most applications of agentic LLMs, there is no formal verification system that can automatically identify the errors in the output of an LLM. In ATP, however, we have the Lean compiler which automatically provides the least of errors, their exact locations in the code, and some descriptions about the errors. Hence, we try to leverage the Lean compiler as much as possible in our agentic approach. Like most agentic approaches in the LLM literature, Apollo also takes inspiration from the workflow of humans, in this case, how mathematicians work: break the proof into parts, simplify each piece, and iteratively rebuild until no goals remain. Apollo automates this process by developing a module to perform each of the operations and centers on a single, powerful tool: the Lean language and its built-in solvers. Such design bridges the gap between LLM-driven proof generation and automatic theorem provers.
>
> ### **5. Using tree search strategies for token budget reduction**
>
> We thank the reviewer for the insightful suggestion of using search methods to further enhance the efficiency of the proposed framework. If we could assign a likelihood to each decomposed sub-lemma’s difficulty/correctness, using tree search methods could result in reduction of tokens with no accuracy changes for Apollo. In other words, such a strategy may further reduce the number of tokens used by Apollo without changing the ultimate accuracy we have reported in the paper. Consider a scenario with the three “sorry” statements in a theorem. Indeed, if the first “sorry” is particularly challenging, DF or BF search strategies could reduce token usage by pruning dependent sub-blocks early. In tightly constrained GPU settings, where we solve sub-goals sequentially, this might lower overall token counts. However, our current design emphasizes parallel computation across multiple GPUs, so Apollo tackles all three “sorry” statements simultaneously. This approach may increase the number of generation attempts and total tokens generated, but it yields faster inference when we have three free GPUs that we can utilize in parallel. It’s a system-design tradeoff, and we agree that in single-GPU environments, one could introduce additional search heuristics to guide recursion more intelligently and further reduce token budgets. It could be an interesting future direction for agentic repair systems like Apollo.
>
> ### **6. Additional reproducibility components**
>
> We appreciate the reviewer’s attention to reproducibility. The current demo release was intended to illustrate the core framework rather than serve as the final toolkit. In the full code release, we will include a comprehensive README, a detailed list of installation requirements, prompt templates, and clear instructions or scripts for downloading and preparing all datasets.
>
> ---
>
> We made a significant effort to perform additional experiments and address your feedback. Computing the token budgets demonstrated the advantage of Apollo across the board increasing accuracing and cutting the token counts. We also experimented on new models (Goedel Prover v2) and new datasets (ProofNet and Putnam Bench). Moreover, we made comparisons with MCTS, again demonstrating an advantage. Our paper and its contribution is much clearer as a result. In light of these improvements, we would appreciate it if you consider raising your score. Do you have any remaining questions or concerns?
>
> ---
>
> [1] Taicheng Guo, et al. (2024). Large Language Model Based Multi-agents: A Survey of Progress and Challenges. ĲCAI.
>
> [2] Junde Wu, et al. (2025). Agentic Reasoning: A Streamlined Framework for Enhancing LLM Reasoning with Agentic Tools.
>
> [3] Chen Qian, et al. (2025). Scaling Large Language Model-based Multi-Agent Collaboration. ICLR

---

> > ### Comment · Reviewer_hqN5 · 2025-08-06
> >
> > Thanks for your response. In light of your added explanation, I am willing to increase my score by one.

---

### Note · Authors · 2025-08-13

Dear AC and Reviewers,

We sincerely thank you for your thoughtful and constructive feedback on our work, as well as the positive comments on our formal proof repair framework. The main points raised in the reviews required us to report additional information about our original experiments, e.g., reporting the token budgets, statistics about invocation of each module in the framework, more clear and detailed information about sample budgets, etc. **Our method itself remains unchanged, and the original experimental setup is intact.**

We reported additional experiments on Goedel Prover-v2-8B improving the reported accuracy of this model. Moreover, we reported new experiments on new datasets: ProofNet and Putnam Bench. **All the additional experiments and comparisons, including the experiments on tree search methods demonstrated an advantage for our proposed method. In all cases, the token budget was reduced and accuracy was improved.**

Comparing Apollo with an iterative self-correction method, raised by the reviewer KutC in their last comment, seems to imply comparison with the method released by Goedel Prover v2 earlier this August. Whether such an iterative method has an advantage over our recursive method remains to be seen. Currently, self-correction behavior is induced via training in Lean theorem provers; however, Apollo does not require such fine-tuning and it still improves the accuracy of the Goedel Prover v2 8B model by 1.6%. Moreover, we note that our paper and method preceded that work by more than three months, and at the time of our submission to NeurIPS, models with a self-correction mode were not yet available.

We appreciate the reviewers’ insightful questions, which have helped us present Apollo’s efficiency, robustness, and generalizability more clearly, even for a model released after the framework. We thank you again for your time and consideration, and we believe our paper is now substantially stronger and more well-rounded.

Sincerely,

Submission #7984 Authors

---

### Decision · Program_Chairs · 2025-09-17

**Decision:**

Accept (poster)

**Comment:**

I recommend accepting this paper. APOLLO combines Lean compiler feedback with LLM reasoning to improve automated theorem proving, achieving 75% accuracy on miniF2F while reducing sampling costs, and boosting general-purpose models from 3-7% to over 40% accuracy. Reviewers (scores: 3, 4, 5) recognized its valuable contribution in integrating compiler feedback effectively across different models, demonstrating how targeted repair can significantly improve formal reasoning efficiency.